# Functional and molecular heterogeneity of D2R neurons along dorsal ventral axis in the striatum

Emma Puighermanal[1,2✉], Laia Castell[1], Anna Esteve-Codina [3], Su Melser[4,5], Konstantin Kaganovsky [6], Charleine Zussy[1], Jihane Boubaker-Vitre[1], Marta Gut[3,7], Stephanie Rialle[1], Christoph Kellendonk[8,9], Elisenda Sanz[2], Albert Quintana [2], Giovanni Marsicano[4,5], Miquel Martin[1], Marcelo Rubinstein[10,11,12], Jean-Antoine Girault [13,14,15], Jun B. Ding[6] & Emmanuel Valjent [1✉]

Action control is a key brain function determining the survival of animals in their environment. In mammals, neurons expressing dopamine D2 receptors (D2R) in the dorsal striatum (DS) and the nucleus accumbens (Acb) jointly but differentially contribute to the fine regulation of movement. However, their region-specific molecular features are presently unknown. By combining RNAseq of striatal D2R neurons and histological analyses, we identified hundreds of novel region-specific molecular markers, which may serve as tools to target selective subpopulations. As a proof of concept, we characterized the molecular identity of a subcircuit defined by WFS1 neurons and evaluated multiple behavioral tasks after its temporally-controlled deletion of D2R. Consequently, conditional D2R knockout mice displayed a significant reduction in digging behavior and an exacerbated hyperlocomotor response to amphetamine. Thus, targeted molecular analyses reveal an unforeseen heterogeneity in D2R-expressing striatal neuronal populations, underlying specific D2R's functional features in the control of specific motor behaviors.

[1] IGF, CNRS, INSERM, Université Montpellier, Montpellier, France. [2] Neuroscience Institute, Department of Cell Biology, Physiology and Immunology, Autonomous University of Barcelona, Bellaterra, Spain. [3] Centre for Genomic Regulation, Barcelona Institute of Science and Technology, 08028 Barcelona, Spain. [4] INSERM U1215, Neurocentre Magendie, Bordeaux, France. [5] Neurocentre Magendie, Université de Bordeaux, Bordeaux, France. [6] Department of Neurosurgery and Department of Neurology and Neurological Sciences, Stanford University School of Medicine, Palo Alto, CA, USA. [7] Department of Experimental and Health Sciences, Universitat Pompeu Fabra, 08003 Barcelona, Spain. [8] Departement of Psychology and Pharmacology, Columbia University, New York, NY, USA. [9] Division of Molecular Therapeutics, New York State Psychiatric Institute, New York, NY, USA. [10] Instituto de Investigaciones en Ingeniería Genética y Biología Molecular, CONICET, Buenos Aires, Argentina. [11] FCEN, Universidad de Buenos Aires, Buenos Aires, Argentina. [12] Department of Molecular and Integrative Physiology, University of Michigan Medical School, Ann Arbor, MI, USA. [13] INSERM UMR-S 1270, Paris, France. [14] Faculty of Sciences, Sorbonne University, Paris, France. [15] Institut du Fer à Moulin, Paris, France. ✉email: emma.puighermanal@gmail.com; emmanuel.valjent@igf.cnrs.fr

The striatum is the gateway to the basal ganglia, an ensemble of subcortical structures involved in motor planning and action selection[1]. Striatal dysfunction has been associated with multiple neurological and psychiatric disorders, including Parkinson's and Huntington's disease, Tourette's syndrome, schizophrenia, autism, and addiction[2,3].

In the striatum, dopamine D2 receptors (D2R) have been tightly linked to a wide variety of motor- and reward/aversion-related behaviors. Pharmacological and genetic studies have demonstrated a direct involvement of D2R neurons in a wide range of functions including motor control[4–6], aversive learning[7], addiction[8], compulsive food-intake[9], motivational aspects of chronic pain[10], and risky decision-making[11]. However, many of these studies used optogenetic or chemogenetic approaches to manipulate D2R neurons, but they did not assess the function of D2R itself in these neurons. Imaging studies in humans and rats have shown a significant reduction in striatal D2R availability in subjects addicted to drugs such as cocaine, alcohol, heroin, nicotine, and methamphetamine as well as in obesity[12]. However, the widespread expression of striatal D2R—which are present in indirect pathway striatal projection neurons (iSPNs), cholinergic interneurons (CINs), and presynaptically in both dopaminergic and glutamatergic afferents—have impeded the interpretation of many pharmacological and behavioral experiments. Our understanding of the complexity of striatal D2R expression pattern and its associated function has recently increased with a study reporting a relative heterogeneity of iSPN subpopulations[13]. In addition, the majority of studies that aimed to uncover the role of D2R, by using either global or conditional D2R knockout mice, could not rule out developmental compensatory adaptations from deleting D2R early in life, since their genetic approaches were not temporally controlled.

Given that the rodent striatum is divided into two regions, the dorsal striatum (DS) and the nucleus accumbens (Acb), which have distinct input–output organization and play different roles in behavior[1,14], we first aimed to identify DS- and Acb-specific molecular markers, which may serve as tools to assess the role of D2R in striatal subpopulations in adult mice. Therefore, we performed RNAseq in DS and Acb D2R neurons separately followed by histological analyses and revealed hundreds of region-specific markers. As a proof of concept, we focused on a cell subpopulation, identified by the expression of Wolfram syndrome 1 (*Wfs1*), displaying a specific expression pattern within the Acb. Following the characterization of the molecular identity of WFS1 neurons, we evaluated the behavioral effect of temporally controlled deletion of D2R from this neuronal subpopulation (*Wfs1-CreERT2:Drd2^loxP/loxP* mice, hereafter named D2R-cKO). We found that D2R-cKO mice displayed altered digging behavior and an exacerbated hyperlocomotor response to amphetamine. Together, our cell type- and region-specific high-throughput analyses uncover previously unknown molecularly and functionally defined subpopulations of D2R neurons and hence reveal novel striatal subcircuits. As a proof of principle, the deletion of D2R from one of these subpopulations identified, WFS1 SPNs, revealed a novel D2R's role in an innate behavior as well as in response to a drug of abuse.

## Results

### Uncovering region-specific molecular markers of striatal D2R neurons.
To identify genes that are preferentially expressed in DS and Acb D2R neurons, we generated *D2-RiboTag* mice[15], which express the Cre-dependent ribosomal protein rpl22 tagged with the hemagglutinin (HA) epitope exclusively in D2R cells (Fig. 1a, b). HA expression —which was selective to iSPNs and CINs[16], and homogeneously spread in both DS and Acb (Fig. 1a, b and

Supplementary Fig. 1a)— enabled the immunoprecipitation of ribosome-bound associated mRNAs selectively from D2R cells. To validate the specificity of the mouse line, we used quantitative reverse transcription PCR (qRT-PCR) to compare the relative abundance of transcripts after HA-immunoprecipitation on whole striatal extracts and the input fraction that contains transcripts from all cell types (Supplementary Fig. 1b). As expected, expression of iSPN markers (*Drd2*, *Adora2*, *Penk1*) was enriched after HA-immunoprecipitation (Fig. 1b). By contrast, gene expression of markers for direct pathway SPNs (dSPNs) (*Drd1*, *Pdyn*, *Tac1*), astrocytes (*Gfap*), microglia (*Aif1*), oligodendrocytes (*Cnp*), as well as GABAergic interneuron-specific markers (*Sst*, *Calb2*, *Pvalb*, *Npy*) were all decreased. No differences were observed for *Chat* (only a fraction of CINs express D2R, see below) and *Th* transcripts (Fig. 1b).

To assess the overlap of DS and Acb D2R translatome profiles, we performed high-throughput RNAseq of tagged ribosome-bound mRNAs following dissection of the two regions (Supplementary Fig. 1a and Supplementary Data 1). Principal component analysis (PCA) revealed that mRNAs' origin (ribosome-bound vs total inputs) represented the main source of variance (81%), while the origin of mRNA (DS vs Acb) accounted for 12% of total variation in the data (Supplementary Fig. 2a). PCA showed that the three replicates of each condition clustered together. Moreover, DS and Acb groups as well as the input and pellet fractions were all well separated between them. Heatmap of sample-to-sample distances confirmed that data were highly reproducible and biological samples had low variability (Supplementary Fig. 2b). We first conducted parallel RNAseq of the inputs (supernatant fraction containing mRNAs from all cell types) and pellets (immunoprecipitation fraction containing tagged ribosomes-bound mRNAs) to elucidate the genes that were enriched in DS and Acb D2R cells (Fig. 1c–f and Supplementary Data 2). Using an adjusted *p* value of < 0.05, our analysis identified 6201 D2R Acb- and 6253 D2R DS-enriched protein-coding genes compared with Acb and DS inputs, respectively (Fig. 1c, e, Supplementary Fig. 3 and Supplementary Data 2). Filtering these genes for a fold-change > 1.5, we narrowed down this list of candidates to 2315 for the Acb and 2260 for the DS (Fig. 1d, f and Supplementary Data 2). Comparison of our analysis with previous results from single-cell RNAseq data[13] showed a match of ~80% of D2R-enriched mRNAs (Fig. 1g and Supplementary Data 2). After filtering for redundancy, we found that among the 6201 D2R Acb- and the 6253 D2R DS-enriched genes, only 459 and 510 were exclusive to D2R cells from Acb and DS, respectively, (Fig. 1h and Supplementary Data 2). We then broadened our analysis, without prefiltering the pellets with the inputs, to capture all genes with DS–Acb differential expression—regardless of their expression profiles outside of D2R neurons (Fig. 1i, j and Supplementary Data 2). Hierarchical clustering of the top 50 differentially expressed genes clearly separates the Acb and DS (Fig. 1i and Supplementary Data 2). Particularly, we found 2797 and 3884 protein-coding gene products more expressed in the Acb and the DS, respectively, (Fig. 1i, j and Supplementary Data 2).

### Genes preferentially expressed in DS D2R neurons.
To identify molecular markers that are preferentially expressed in the DS, we analyzed genes that showed a significant dorso-ventral expression bias in our high-throughput analysis. We found 3884 protein-coding genes preferentially expressed in the DS (Fig. 1i and Supplementary Data 2). Some of these genes include *Trnp1*, *Lpcat4*, *Kctd17*, *Trpc3*, *Ace*, *Dab2ip*, *Me2*, *Rgs4*, *Itga5*, *Coch*, *Tbc1d8*, *Gpr155*, *Rasd2*, *Rgs7bp*, *Slc24a2*, *Kcnk2*, *Ddit4l*, and *Ccnd2* among others (Fig. 2a). This enrichment was confirmed by

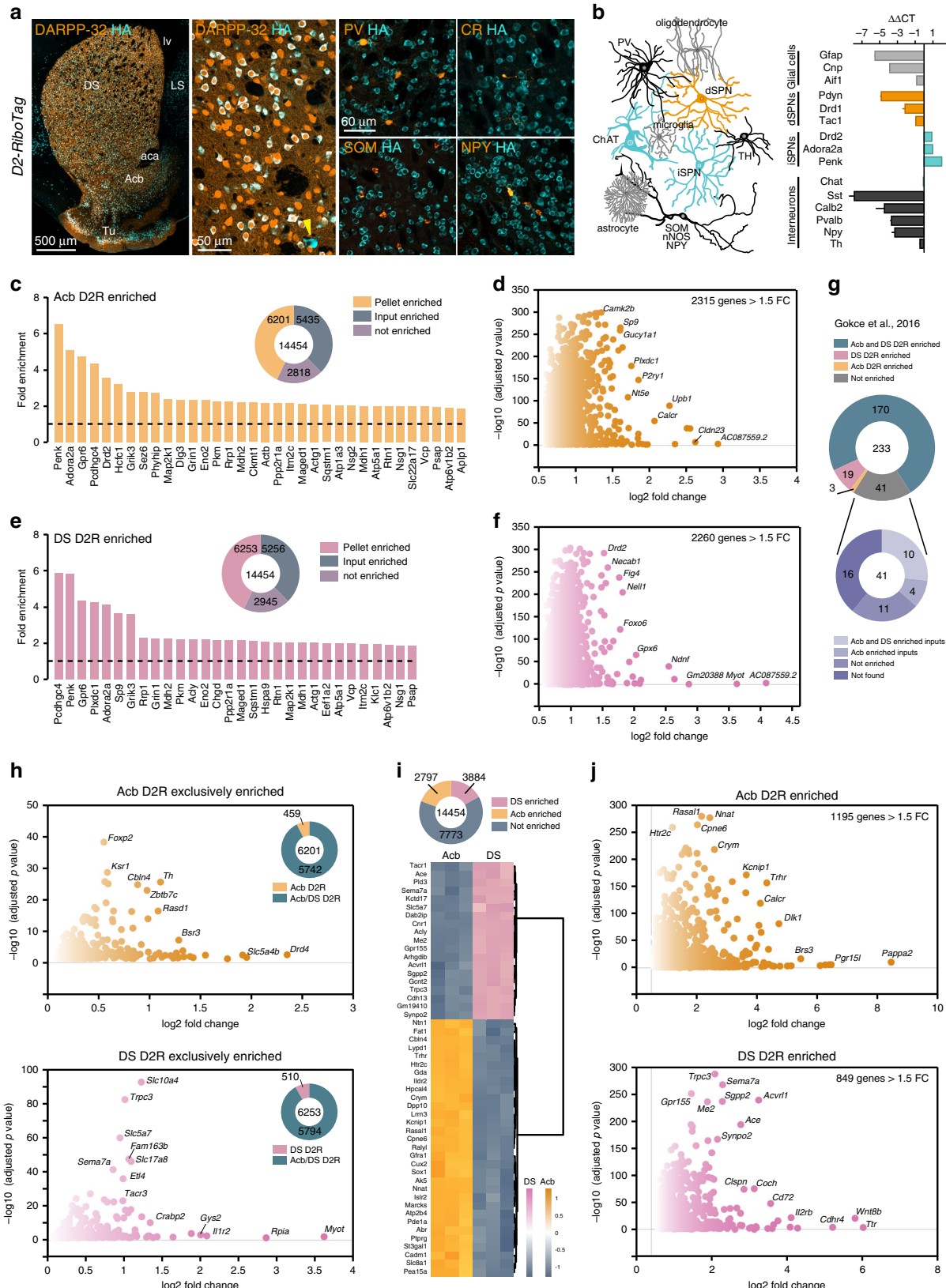

qRT-PCR analysis performed following HA-immunoprecipitation in different *D2-RiboTag* mice (Fig. 2b). Cross analysis of our data with in situ hybridization (ISH) profiles from the Allen Brain Atlas further validated the differential gene expression. Close inspection of the distribution of genes more expressed in DS indicated that expression patterns are highly heterogeneous and could be visually classified into different categories including widespread (*Foxp1, Pcp4, Camkk2, Itpr1*), lateral DS restricted (*Ace, Me2, Rgs7bp, Acvrl1*), medial DS restricted (*Rbp4, Fos, Ntm, Ddit4l*), sparse (*Cd4, Cit, B3gnt2*), or patch/matrix (*Calb1, Pdp1, Cdh8, Plxnd1, Sepw1*)[17] (Fig. 2c and Supplementary Fig. 4a). Differential expression was also confirmed at the protein level for neurogranin

**Fig. 1 Translatome profile of DS and Acb D2R neurons using D2-Ribotag mice. a** Coronal striatal section of *D2-RiboTag* mice stained with HA (cyan) and DARPP-32 (orange), PV (parvalbumin), CR (calretinin), NPY (neuropeptide Y), or SOM (somatostatin). Note the selective HA expression in ~50% of SPNs (DARPP-32-positive) corresponding to iSPNs and in CINs (arrowhead). **b** Drawing summarizing HA expression (cyan) among the distinct striatal cell types and validation by qRT-PCR ($\Delta\Delta CT$) of the enrichment of iSPN markers and de-enrichment of dSPNs, interneurons, and glial cells markers after HA-immunoprecipitation on whole striatal extract (DS and Acb) compared with the input fraction (containing the mRNAs from all cellular types) ($n = 4$ mice/ group). Data are presented as mean values ± SEM. **c** Fold-change of protein-coding genes enriched in the Acb pellet fraction of *D2-RiboTag* mice. **d** Volcano plot depicting protein-coding genes enriched in the Acb of D2R neurons. **e** Fold-change of protein-coding genes enriched in the DS pellet fraction of *D2-RiboTag* mice. **f** Volcano plot depicting protein-coding genes enriched in the DS of D2R neurons. **g** Doughnut chart showing the overlap and distribution of D2R-enriched genes found in our study among the 233 D2R-enriched genes identified in [13]. **h** Volcano plot depicting protein-coding genes enriched in the Acb (top panel) and in the DS (bottom panel) of D2R neurons after filtering the pellet fraction with the input fraction. **i** Heatmap of the top 50 genes most significantly enriched either in DS (pink) or Acb (orange). Scaled expression values are color coded according to the legend. The dendrogram depicts hierarchical clustering. **j** Volcano plot depicting protein-coding genes from the pellet fraction of D2R neurons that are enriched in the Acb (top panel) and in the DS (bottom panel). DS dorsal striatum, LS lateral septum, Acb accumbens, Tu olfactory tubercules, aca anterior commissure, lv lateral ventricle.

(Nrgn), D2R, calbindin-D28k (CB), and dopamine- and cAMP-regulated phosphoprotein, Mr 32 kDa (DARPP-32) (Fig. 2d).

Notably, our RNAseq approach was highly valuable to detect differential expression levels of genes encoding proteins mainly expressed presynaptically, such as *Cnr1*, the cannabinoid receptor type 1 (CB1R). Western blot (WB) analysis of CB1R expression only revealed a slight increase (~13%) in DS compared with Acb (Fig. 2e), presumably due to the presence of CB1R in striatal afferents and the fact that CB1R is mainly expressed in SPNs terminals (Fig. 2f). In contrast, when analyzing the mRNA level present in iSPNs, both RNAseq and qRT-PCR revealed a 8.6- and 3.3-fold enrichment respectively of CB1R in DS compared with Acb (Fig. 2g).

Our analysis also revealed a remarkable dorso-ventral expression gradient of mitochondria-related genes. This bias is particularly evident for genes encoding for mitochondria ribosomal proteins (17 in DS vs 5 in Acb), mitochondrial transporters (19 in DS vs 4 in Acb), translocases (12 in DS vs 0 in Acb), and respiratory chain complex (69 in DS vs 4 in Acb) (Fig. 2h and Supplementary Data 3). Among the five complexes, bias expression was particularly unbalanced since 26 out of 44 genes encoding for proteins of the respiratory chain complex I (CI) are more actively translated in the DS (Fig. 2h and Supplementary Data 3). This bias is functionally relevant since the activity of CI measured in total homogenate was higher in the DS compared with the Acb (Fig. 2i). The citrate synthase (CS) activity, a validated biomarker for mitochondrial density, was also slightly increased in the DS (Fig. 2j), suggesting that mitochondria content might be higher in the DS than in the Acb. However, the increased CI activity was presumably not only due to a higher mitochondrial density since the CI/CS ratio was still significantly increased in the DS (Fig. 2k).

**Genes preferentially expressed in Acb D2R neurons.** Similar to DS, our RNAseq analysis revealed 2797 protein-coding genes that show a preferential expression in the Acb (*Peg10, Calcr, Cbln4, Amotl1, Gabrg1, Ntn1, Stard5, Lrrn3, Wfs1, Nts, Cartpt, Nnat, Hap1, Trhr, Hpcal4, Gfra1, Dlk1, Fam126a*) (Fig. 3a). All these genes were confirmed by qRT-PCR in different *D2-RiboTag* mice (Fig. 3b). Cross analysis with the ISH distribution patterns also revealed that genes more expressed in Acb are widespread (*Slc35d3, Crym, Nnat*) or preferentially distributed in distinct Acb territories including the core (AcbC) (*Cartpt, Hpcal4, Col6a1, Zdbf2, Calcr, Dlk1, Peg10*) as well as the medial (*Hap1, Chn2, Gprin1, Stard5, Lrrn3, Foxp2*), ventral (*Gpr101, Gfra1, Cdh2, Gpr26, Cpne6*), lateral (*Nts, Gpr83*), or cone part (*Trhr, Cpne2, Lypd1, Carhsp1, Dpp10*) of the shell (AcbSh) (Fig. 3c, d, g and Supplementary Fig. 4b). The biased expression toward the Acb was also confirmed at the protein level for Gat1, Gprin1, Foxp2, Map2, Sox1, and Hap1 (Fig. 3d).

Interestingly, we noticed a preferential Acb expression of many imprinted genes, which are those genes whose expression occurs from only one allele and represent < 1% of all genes. Among the 90 imprinted genes detected in the striatum, half of them were differentially expressed between the two regions, with 42 genes more expressed in the Acb and only 19 in the DS (Fig. 3e and Supplementary Data 3). The biased Acb expression of two paternally imprinted genes, Peg10 and Nnat, were confirmed by WB (Fig. 3a, b, f). Strikingly, ISHs revealed distinct localization patterns within the Acb (Fig. 3d, g and Supplementary Fig. 4b), as well as different expression levels compared with other brain regions. For instance, *Calcr, Dlk1, Zdbf2,* and *Peg10* are almost exclusively expressed in the Acb compared with surrounding structures (Fig. 3d, g and Supplementary Fig. 4b), while *Cntn3, Gnas,* and *Nap1l5* expression is rather widely distributed. Of note, several imprinted genes such as *Dlk1, Calcr, Zdbf2,* and *Peg10* show a rather selective expression in the AcbSh cone and ventral part of the AcbC (Fig. 3c, g).

We also found transcripts of the imprinted gene *Th* in Acb D2R neurons (Fig. 3e). Although a low number of reads suggests that *Th* is probably weakly expressed (Supplementary Data 2), scattered expression was detected by ISH in the Acb (Fig. 3h). This pattern matches with the enriched GFP staining observed in the bundle-shaped area of the caudomedial AcbSh in *Th-eGFP* mice (Fig. 3i). Consistent with our RNAseq data, we found that half of the TH neurons also express DARPP-32 in both AcbC and AcbSh (Fig. 3j, k). A minor proportion of cells was also positive for calretinin (CR) but not for NPY or ChAT (Fig. 3l, m). These results suggest that in addition to GABAergic interneurons, some iSPNs also express TH, thereby defining a novel subpopulation of D2R-TH-SPNs.

**Heterogeneity of DS and Acb CINs.** Our RNAseq and qRT-PCR analyses also revealed several CINs-enriched genes, including classical CINs markers (*Chat, AChE, Slc18a3, Slc17a8*) as well as markers previously found by the bacTRAP approach[18] (*Ufsp1, Ecel1, Bves, Crabp2, Ntrk1*). Finally, several additional CINs-enriched genes were also identified (*Slc10a4*[19], *Kctd6, S100a10, Tacr1*[20], *Tacr3*) (Supplementary Fig. 5a, b). The expression of all these DS-enriched genes was confirmed by qRT-PCR in different *D2-RiboTag* mice (Supplementary Fig. 5a, b). Enrichment for VGLuT3, TrkA, and NK1R was also confirmed by triple immunofluorescence (IF) analysis in *D2-eGFP* mice with either VAChT or ChAT (Supplementary Fig. 5c–e). Because CINs are differentially distributed across the mouse striatum[21] (Supplementary Fig. 5c), we determined the percentage of ChAT/HA-positive cells in the DS and Acb (Supplementary Fig. 5f, g). Our analysis revealed a gradual decrease in the fraction of ChAT/HA colabeled cells ranging from 100% co-expression in the DS (360 HA$^+$ neurons out of 360 ChAT$^+$ cells) to 88% in the AcbC

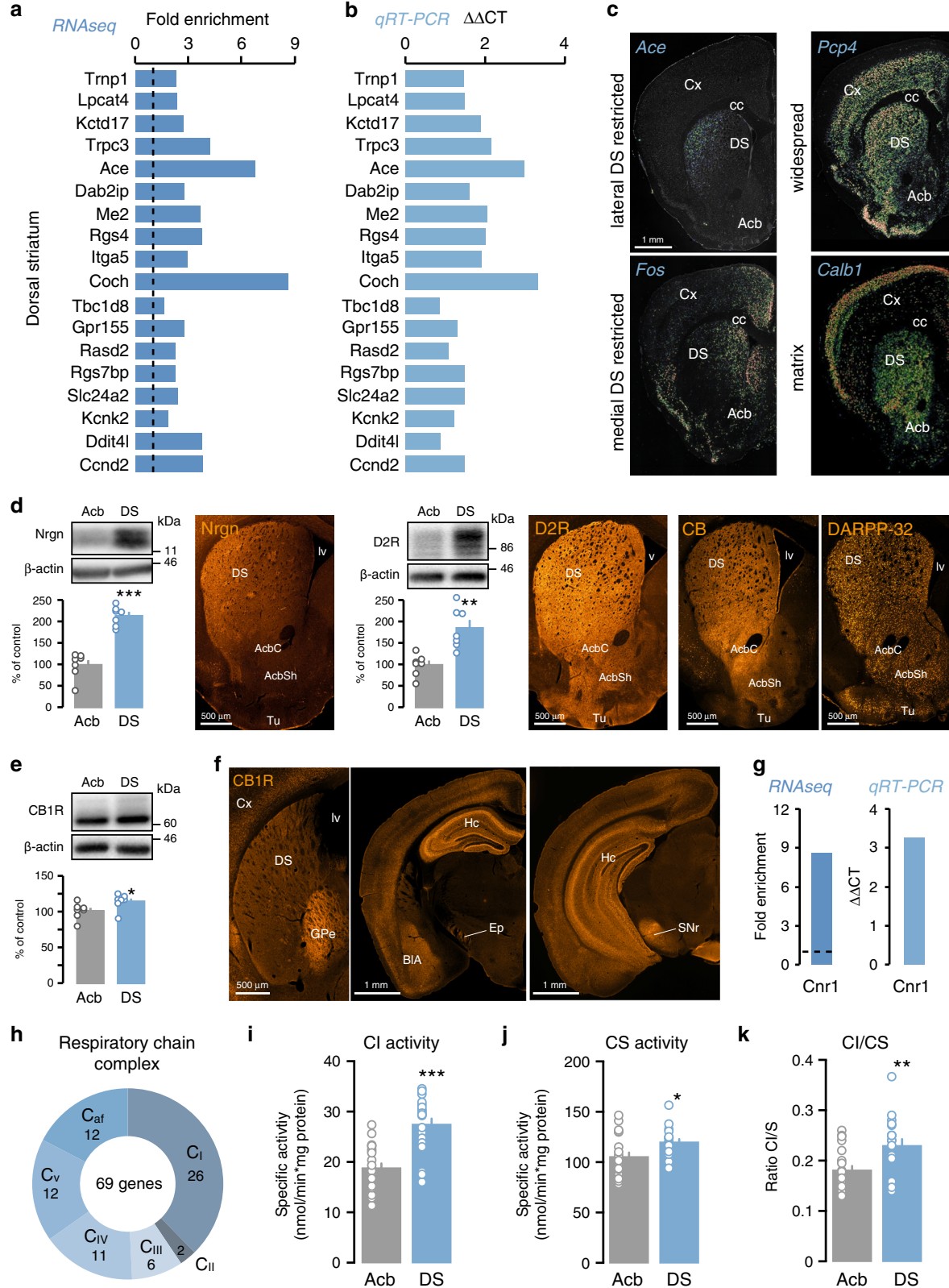

(83 HA⁺ neurons out of 94 ChAT⁺ cells) and only 38% in the AcbSh (36 HA⁺ neurons out of 95 ChAT⁺ cells) (Supplementary Fig. 5f, g). This observation could explain why CINs-enriched genes were preferentially found in the in DS and the lack of ChAT enrichment after HA-immunoprecipitation from whole striatum extract (Fig. 1c). Our cross analysis with recent results obtained by single-cell RNAseq analysis[22] confirmed that ~83%

of their CINs-enriched genes were also found in a population of D2R CINs (Supplementary Fig. 5h and Supplementary Data 4). Interestingly, although less represented, Acb D2R CINs-enriched genes can also be identified (Supplementary Fig. 5h and Supplementary Data 4). Thus, similar to the heterogeneity of iSPNs, these results indicate that numerous CINs subpopulations may exist.

**Fig. 2 Identification of genes from D2R cells that have a preferential expression in the DS.** Fold-change and $\Delta\Delta CT$ of DS-enriched genes found by RNAseq (**a**) and confirmed by qRT-PCR (**b**). Different cohorts of *D2-RiboTag* mice were used for both techniques. **c** ISH coronal sections from Allen Brain Atlas of region-specific DS-enriched genes. **d** Enrichment of Nrgn ($t_{12} = 8.106$, $p = 0.000003$, two-sided $t$ test, $n = 7$ mice/group), D2R ($t_{12} = 4.24$, $p = 0.0011$, two-sided $t$ test, $n = 7$ mice/group), CB, and DARPP-32 in the DS confirmed by WB and IF analyses ($n = 3$ mice/group). **e** WB and quantification of CB1R in DS and Acb ($t_{12} = 2.222$, $p = 0.0463$, two-sided $t$ test, $n = 7$ mice/group). **f** CB1R staining in the DS and its output structures ($n = 4$ mice/group). **g** RNAseq fold-change and $\Delta\Delta CT$ of qRT-PCR of *Cnr1*. **h** Doughnut chart showing the distribution of the DS D2R-enriched genes (69) belonging to the respiratory chain complex I ($C_I$, 26), II ($C_{II}$, 2), III ($C_{III}$, 6), IV ($C_{IV}$, 11) and complex V ($C_V$, 12) and to the respiratory chain complex assembly factors ($C_{af}$, 12). **i** Enzymatic activity of complex I (CI) in Acb or DS ($t_{41} = 5.791$, $p = 0.000013$, two-sided $t$ test, $n = 22$ mice/Acb and $n = 21$ mice/DS). **j** Enzymatic activity of citrate synthase (CS) in Acb or DS ($t_{39} = 2.706$, $p = 0.0101$, two-sided $t$ test, $n = 21$ mice/Acb and $n = 20$ mice/DS). **k** Ratio CI/CS in Acb or DS ($t_{38} = 3.109$, $p = 0.0035$, two-sided $t$ test, $n = 21$ mice/Acb and $n = 19$ mice/DS). All data are presented as mean values ± SEM. DS dorsal striatum, Cx cortex, LS lateral septum, Acb accumbens, AcbC accumbens core, AcbSh accumbens shell, GPe external globus pallidus, Hc hippocampus, Ep entopeduncular nucleus, BlA basolateral amygdala, SNr substantia nigra part reticulata, Tu olfactory tubercules, aca anterior commissure, lv lateral ventricle, cc corpus callosum.

**Systematic molecular classification of DS and Acb D2R neurons.** In order to provide further information about molecular expression biases between the DS and the Acb, we implemented a classification by sorting genes of interest according to neurotransmitter systems based on the publicly available IUPHAR/BPS database (www.guidetopharmacology.org). Thus, genes encoding receptors, transporters, and enzymes involved in neurotransmitter turnover were categorized as Acb-enriched (orange), DS-enriched (pink), not enriched (gray), or not expressed (black) and organized for various neurotransmitters systems including 5-HT, catecholamines, GABA, glutamate acetylcholine, and endocannabinoid (Supplementary Fig. 6a–p). This representation, which allows a rapid overview of the regional enrichment of genes in D2R neurons, revealed overwhelming differences between the DS and Acb. A bias toward the Acb is particularly evident in expression of 5-HTRs (*Htr1a*, *Htr2a*, *Htr2c*, *Htr4*, and *Htr7*) (Supplementary Fig. 6a–d) and for GABAa receptor subunits (Supplementary Fig. 6h–j).

Remarkable region-specific differences were also found in the catecholaminergic system. For example, the gene encoding D2R is highly enriched in the DS (Supplementary Fig. 6f), supporting the bias shown earlier by WB and IF (Fig. 2d). However, these protein-level analyses also included D2R from cortico- and thalamostriatal, and dopaminergic terminals, and were not exclusive from iSPNs and CINs as in our RNA-based approach. Preferential expression of *Drd5* was also observed in the DS most likely corresponding to that found for CINs[23] (Supplementary Fig. 6f and Supplementary Data 4). Our approach also revealed enrichment of *Drd1* and *Drd3* mRNAs in the Acb (Supplementary Fig. 6f), indicating that these DA receptors might also be present in D2R neurons, as supported by the significant number of GFP/RFP-positive cells found in AcbSh of *D2-eGFP/D1-tdTomato* mice (Supplementary Fig. 6g). This enrichment of *Drd1* transcripts in Acb D2R neurons provides additional evidence for a higher proportion of D1R/D2R neurons in the Acb[24–28].

Similar classification was performed for voltage-gated ion channels including calcium, sodium, and potassium channels, which comprise inwardly rectifying potassium channels, voltage-gated potassium channels, two-P potassium channels, calcium-activated potassium channels, and accessory subunits (Supplementary Fig. 7a–g). Future studies may demonstrate a relationship between the intrinsic electrophysiological properties of D2R subpopulations and channel expression levels. Finally, Gene Ontology (GO) analysis revealed an important dichotomy between genes more expressed in DS and Acb that might underlie diversity in the biological functions of DS and Acb D2R neurons (Supplementary Data 5–7).

**Characterization of Acb WFS1 neurons.** To illustrate how the identification of hundreds of novel region-specific molecular markers may serve as tools to parse the role of D2R in selective striatal subpopulations, we focused as a proof of concept on Acb WFS1 neurons. We selected this cell subpopulation based on the availability of genetic tools and because the *Wfs1* gene displayed one of the most segregated expression patterns between the DS and Acb. Both ISH and IF analyses revealed a prominent expression in the Acb, especially in the AcbC and intermediate AcbSh, while weaker in the lateral and medial AcbSh (Fig. 4a, b). To gain insight into the molecular identity of this subpopulation, we generated *Wfs1-Ribotag* mice (Fig. 4c) and performed RNAseq of the tagged ribosome-bound mRNAs and the input fractions of the Acb. The specificity of the approach was validated since *Wfs1* mRNA was enriched in the pellet fraction, while transcripts of markers of astrocytes (*Gjb6*, *Kncj10*, *Slc1a2*), oligodendrocytes (*Olig2*, *Cnp*, *Clcn11*), microglia (*Aif1*, *Itgam*, *Tmem119*), and vascular cells (*Flt1*, *Gata2*, *Sox18*) were depleted (Fig. 4d). Our analysis ($p < 0.05$) identified 5885 protein-coding genes enriched in WFS1 neurons (Fig. 4e, f and Supplementary Data 8).

We next examined the presence of markers of specific striatal cell types within the WFS1 subpopulation. Among the genes enriched in WFS1 neurons, we found transcripts expressed in SPNs (*Ppp1r1b*, *Bcl11b*, *Gpr88*, *Rgs9*), including both dSPNs (*Drd1*, *Pdyn*, *Tac1*, *Chrm4*) and iSPNs (*Drd2*, *Adora2a*, *Penk*, *Gpr6*) (Fig. 4g and Supplementary Data 8). These results were confirmed after analyzing the distribution of WFS1 in both D1R- and D2R-SPNs (Fig. 4h, i). While WFS1 appears to be particularly enriched in D1R AcbC and AcbSh neurons (Fig. 4h, i), it is also expressed by Acb D2R neurons (Fig. 4g–j). By contrast, no expression of *Wfs1* was detected in CINs or other types of interneurons (Fig. 4k–m). Since the Acb subcircuitry defined by WFS1 cells has never been investigated, we next assessed the projection patterns of WFS1 neurons in known target structures of the Acb. To do this, the Cre-inducible viral vector AAV2/5-hSyn-DIO-rM3D(Gs)-mCherry was injected into the Acb (injection covering the AcbC and AcbSh lateral) of *Wfs1-CreERT2* mice and RFP (mCherry) immunoreactivity was analyzed in Acb output structures (Fig. 4n, o). RFP staining was found in the ventral pallidum (VP) and SN/VTA identified with enkephalin and TH labeling, respectively, (Fig. 4o).

As revealed by PCA, the enriched transcripts found in Acb D2R neurons markedly differed from the WFS1 subpopulation (Supplementary Fig. 2). We therefore compared the highest differentially expressed genes in Acb WFS1- and D2R-positive neurons (Fig. 5). This analysis revealed that only 443 out of the 5885 were selectively enriched in WFS1 neurons (Fig. 5a and Supplementary Data 9). Conversely, 758 out of the 6201 enriched genes found in Acb D2R neurons were absent from the genes enriched in WFS1 neurons (Fig. 5b and Supplementary Data 9). When taking into account only the enriched transcripts from Acb D2R neurons after filtering the pellet with the input fraction,

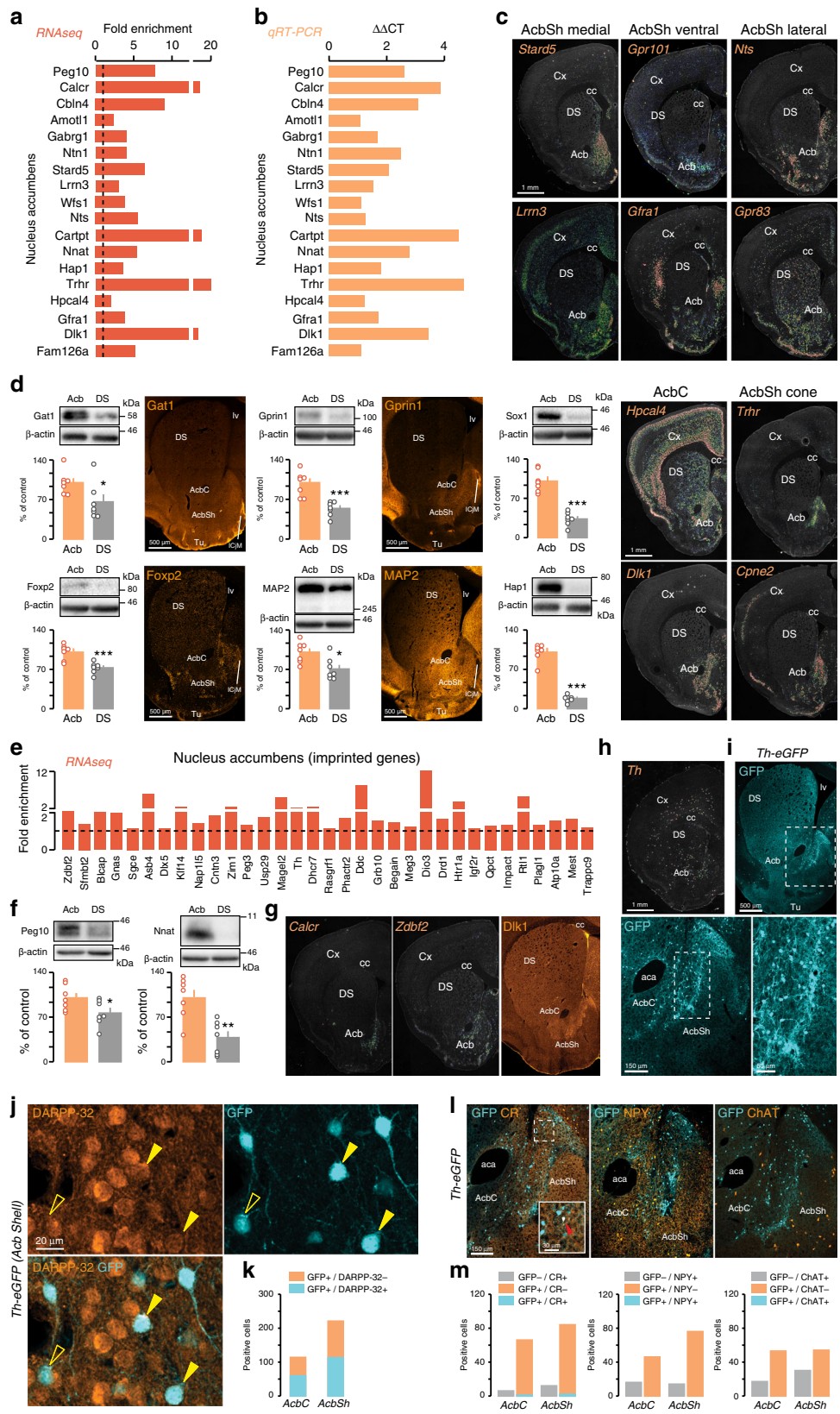

253 out of the 459 transcripts enriched in D2R cells were also present in WFS1 cells, and only 206 were exclusive of D2R cells (Fig. 5c and Supplementary Data 9). A nonfiltered analysis revealed that 2779 and 2447 protein-coding genes were found to be enriched in Acb WFS1- and D2R-positive neurons, respectively, (Fig. 5d and Supplementary Data 9). A classification based

on neurotransmitter systems as well as GO analysis suggested major differences between Acb WFS1 and D2R neurons (Supplementary Figs. 8 and 9 and Supplementary Data 9). Indeed, a bias toward the Acb D2R neurons is particularly evident for subunits of voltage-gated calcium channels, sodium channels, and voltage-gated potassium channels (Supplementary Fig. 9a, b,

**Fig. 3 Identification of genes from D2R cells that have a preferential expression in the Acb.** Fold-change and $\Delta\Delta$CT of Acb-enriched genes found by RNAseq (**a**) and confirmed by qRT-PCR (**b**). Different cohorts of *D2-RiboTag* mice were used for both techniques. **c** ISH coronal sections from Allen Brain Atlas of region-specific Acb-enriched genes. **d** Enrichment of Gat1 ($t_{12} = 2.316$, $p = 0.0390$, two-sided *t* test, $n = 7$ mice/group), Gprin1 ($t_{12} = 5.252$, $p = 0.0002$, two-sided *t* test, $n = 7$ mice/group), Sox1 ($t_{12} = 7.894$, $p = 0.000004$, two-sided *t* test, $n = 7$ mice/group), Foxp2 ($t_{12} = 4.533$, $p = 0.0007$, two-sided *t* test, $n = 7$ mice/group), MAP2 ($t_{12} = 2.892$, $p = 0.0135$, two-sided *t* test, $n = 7$ mice/group), and Hap1 ($t_{12} = 12.98$, $p = 0.00000002$, two-sided *t* test, $n = 7$ mice/group) in the Acb confirmed by WB and IF analyses ($n = 3$ mice/group). All data are presented as mean values ± SEM. **e** Fold-change of statistically significant imprinted genes found by RNAseq after HA-immunoprecipitation in *D2-RiboTag* mice. **f** WB and quantification of Peg10 ($t_{12} = 2.183$, $p = 0.0496$, two-sided *t* test, $n = 7$ mice/group) and Nnat ($t_{12} = 3.97$, $p = 0.0019$, two-sided *t* test, $n = 7$ mice/group) in Acb and DS. All data are presented as mean values ± SEM. **g** ISH of imprinted genes from the Allen Brain Atlas and coronal section of Dlk1 immunostaining ($n = 3$ mice/group). **h** *Th* ISH from the Allen Brain Atlas. **i** Coronal section of GFP staining in *Th-eGFP* mice ($n = 3$ mice/group). **j, k** Double IF for GFP and DARPP-32 and quantification of GFP/DARPP-32 cells in the Acb of *Th-eGFP* mice ($n = 3$ mice/group). **l, m** Double IF for GFP and CR, NPY, and ChAT and quantification of co-expressing cells in the Acb of *Th-eGFP* mice ($n = 3$ mice/group). DS dorsal striatum, Cx cortex, Acb accumbens, AcbC accumbens core, AcbSh accumbens shell, ICjM island of Calleja, Tu olfactory tubercules, aca anterior commissure, lv lateral ventricle, cc corpus callosum.

d) while subunits of inwardly rectifying potassium channels are more expressed in Acb WFS1 neurons (Supplementary Fig. 9c). GO analysis indicated that enriched transcripts from Acb WFS1 neurons are mostly related to the extracellular matrix including cell-adhesion molecules (Fig. 5e, f and Supplementary Data 9). Finally, we cross-analyzed our results with recent single-cell RNAseq study defining new D2R- and D1R-SPNs subpopulations[13]. Among the 61 gene products defining the D2R-Htr7-SPNs subpopulation, 30 were found to be also enriched in Acb D2R neurons in our dataset (Fig. 5g and Supplementary Data 10). Such analysis also confirmed that WFS1 neurons are enriched in gene products classifying D1R-SPNs and some of their discrete subpopulations (Fig. 5h and Supplementary Data 10).

**D2R from WFS1 neurons control digging behavior.** To parse the role of D2R in WFS1 neurons, we generated temporally controlled conditional *Drd2* knock-out mice (D2R-cKO) by crossing the tamoxifen-inducible *Wfs1-CreERT2* BAC transgenic mouse line with the *Drd2^loxP/loxP* line. Three weeks after tamoxifen treatment, the efficiency of *Drd2* ablation was confirmed by ISH using Basescope assay (Fig. 6a). D2R expression analyzed by WB and IF also confirmed that D2R levels were reduced by ~32% in the Acb of D2R-cKO mice (Fig. 6b, c) and that deletion of *Drd2* was restricted to the Acb since no differences in D2R expression were found in the DS between genotypes (Fig. 6b, c). The ~70% preserved D2R expression in the Acb of D2R-cKO mice presumably corresponds to the presynaptic D2R present in dopaminergic and glutamatergic afferents (including both cortico- and thalamostriatal inputs) as well as the postsynaptic D2R expressed in ~50% of Acb iSPNs that do not express *Wfs1* (see Fig. 4h, i).

We first tested whether temporally inducible deletion of *Drd2* in a restricted Acb subcircuit impacted locomotor responses. No differences in total distance traveled or movement speed were found in an open field arena between D2R-cKO and control littermates (Fig. 6d). Similar results were found in horizontal and vertical locomotor activity measured in a circular corridor (Fig. 6e, f). D2R-cKO also displayed similar motor performance on the rotarod as compared with control mice (Fig. 6g). No alterations in anxiety-like and depressive-like behaviors were observed in the elevated plus maze and tail suspension test, respectively, (Supplementary Fig. 10a, b). Moreover, no deficits in working memory or perseverative behavior were found in the spontaneous alternation task (Supplementary Fig. 10c). Intriguingly, D2R-cKO mice exhibited a pronounced impairment in the marble burying test, hiding less than half of the marbles compared with control littermates (Fig. 6h, Supplementary Videos 1 and 2). This phenotype was not due to an effect of novelty because similar results were found by repeating the test the following day, when marbles were already familiar to mice

(Fig. 6h). In addition, no differences were found in the novel object exploration task (Supplementary Fig. 10d) or in the time mice spent exploring the object and stranger in the three-chamber test (Supplementary Fig. 10e, f). Since the phenotype observed in marble burying cannot be explained by alterations in locomotion (Fig. 6d–f), anxiety (Supplementary Fig. 10a), a lack of novelty-seeking behavior (Supplementary Fig. 10d), or an effect of marbles-induced anxiety (Fig. 6h, day 2), we hypothesized that an alteration of innate behaviors such as digging could account for the decrease of marbles buried. Indeed, D2R-cKO mice spent significantly less time digging than control mice (Fig. 6i), despite having a similar latency to start digging and a total number of digging bouts (Supplementary Fig. 10g). Importantly, D2R-cKO mice displayed similar goal-directed digging toward standard or palatable food than control mice (Supplementary Fig. 10h). Finally, this phenotype is not a consequence of a decrease in repetitive behaviors since no differences in grooming behavior (Supplementary Fig. 10i) or in repetitive motor routines[29] (Fig. 6g) were observed between genotypes.

In addition to the role of D2R from WFS1-expressing cells in digging behavior, we sought to assess whether the whole WFS1-positive subcircuit (including both dSPNs and iSPNs) participated in this innate behavior. To do so, we chemogenetically activated Acb WFS1 neurons by bilaterally injecting the Cre-inducible viral vector AAV-hSyn-DIO-rM3D(Gq)-mCherry into the Acb of *Wfs1-CreERT2* mice and systemically injecting its ligand CNO (1 mg/kg). No differences were found between CNO- and vehicle-treated mice in digging behavior, locomotor activity, perseverative behavior, working memory, novel object exploration, or grooming behavior (Supplementary Fig. 11). The lack of effect in digging behavior after the activation of Acb WFS1 neurons, compared with the key role that D2R have in the same subpopulation, could be explained by the simultaneous activation of both direct and indirect pathways that might have opposite roles.

**Deletion of D2R from WFS1 neurons enhances the response to amphetamine.** Next, we assessed the contribution of D2R from WFS1 neurons in response to pharmacological stimulations. Since D2R is the main receptor targeted by antipsychotic drugs, we first evaluated the cataleptic response induced by the typical antipsychotic haloperidol in D2R-cKO mice. In contrast to the blunted haloperidol-induced catalepsy observed in mice lacking the D2L isoform[30,31]—which is mainly expressed at postsynaptic sites—or bearing a selective deletion of D2R in CINs[32], similar cataleptic responses were observed in D2R-cKO mice and control littermates (Fig. 6j). These results indicate that the catalepsy induced by haloperidol does not require D2R in the WFS1 neuronal population.

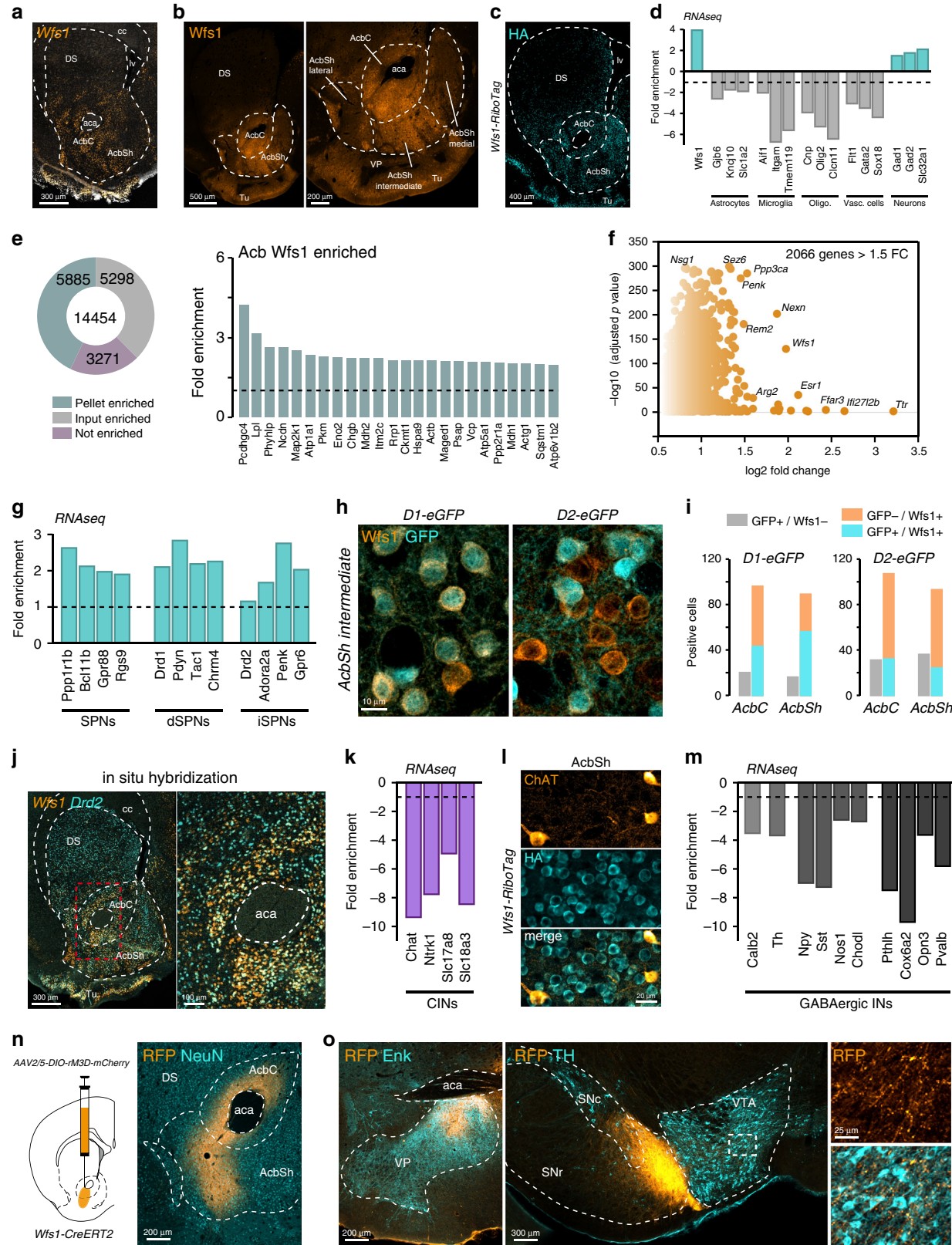

We then stimulated D2R indirectly by boosting striatal DA levels using the psychostimulant amphetamine. After a 3-day habituation with saline injections in a circular corridor, mice received a single injection of amphetamine (2.5 mg/kg), which elicited a robust increase in locomotion in control mice (Fig. 6k). Interestingly, the amphetamine-induced locomotor response was significantly higher in D2R-cKO mice (Fig. 6k), suggesting that D2R in WFS1 neurons negatively control the effects of amphetamine on locomotion. On the other hand, behavioral sensitization was comparable between genotypes as determined by similar slopes and ratios between day 1 and 12 of amphetamine treatment (Fig. 6l). Finally, similar hyperlocomotor

**Fig. 4 Anatomical characterization of Acb WFS1 neurons. a** Single-molecular fluorescent in situ hybridization for *Wfs1* mRNAs in the striatum. **b** Coronal section with Wfs1 immunostaining (*n* = 3 mice/group). **c** Coronal section with HA immunostaining from *Wfs1-RiboTag* mice (*n* = 4 mice/group). **d** Validation by RNAseq of the de-enrichment of markers for astrocytes, microglia, oligodendrocytes, and vascular cells, and the enrichment of GABAergic cells markers after HA-immunoprecipitation on Acb extracts compared with the input fraction (containing the mRNAs from all cellular types). **e** Fold-change of protein-coding genes enriched in the Acb pellet fraction of *Wfs1-RiboTag* mice. **f** Volcano plot depicting protein-coding genes enriched in the Acb of Wfs1 neurons. **g** Fold-change of SPNs, dSPNs, and iSPNs markers from the Acb pellet fraction of *Wfs1-RiboTag* mice. **h** Double IF for GFP and Wfs1 in *D2-* and *D1-eGFP* mice (*n* = 3 mice/group). **i** Quantification of GFP/Wfs1 cells in the Acb of *D2-* and *D1-eGFP* mice. **j** Single-molecular fluorescent in situ hybridization for *Wfs1* (orange) and *Drd2* (cyan) mRNAs in the striatum. **k** Fold-change of CINs markers de-enriched in the Acb pellet fraction of *Wfs1-RiboTag* mice. **l** Double IF of ChAT (orange) and HA (cyan) in the Acb of *Wfs1-RiboTag* mice. **m** Fold-change of striatal interneuron markers de-enriched in the pellet fraction of *Wfs1-RiboTag* mice. **n** Schematic of Cre-dependent AAV-mCherry Acb injection in *Wfs1-CreERT2* mice and visualization of the mCherry expression (gold color) at the injection site as well as (**o**) in output structures identified by co-staining with ENK (ventral pallidum, VB) and TH (substantia nigra pars compacta, SNc, and ventral tegmental area, VTA). DS dorsal striatum, Cx cortex, Acb accumbens, AcbC accumbens core, AcbSh accumbens shell, ICjM island of Calleja, Tu olfactory tubercules, aca anterior commissure, lv lateral ventricle, cc corpus callosum.

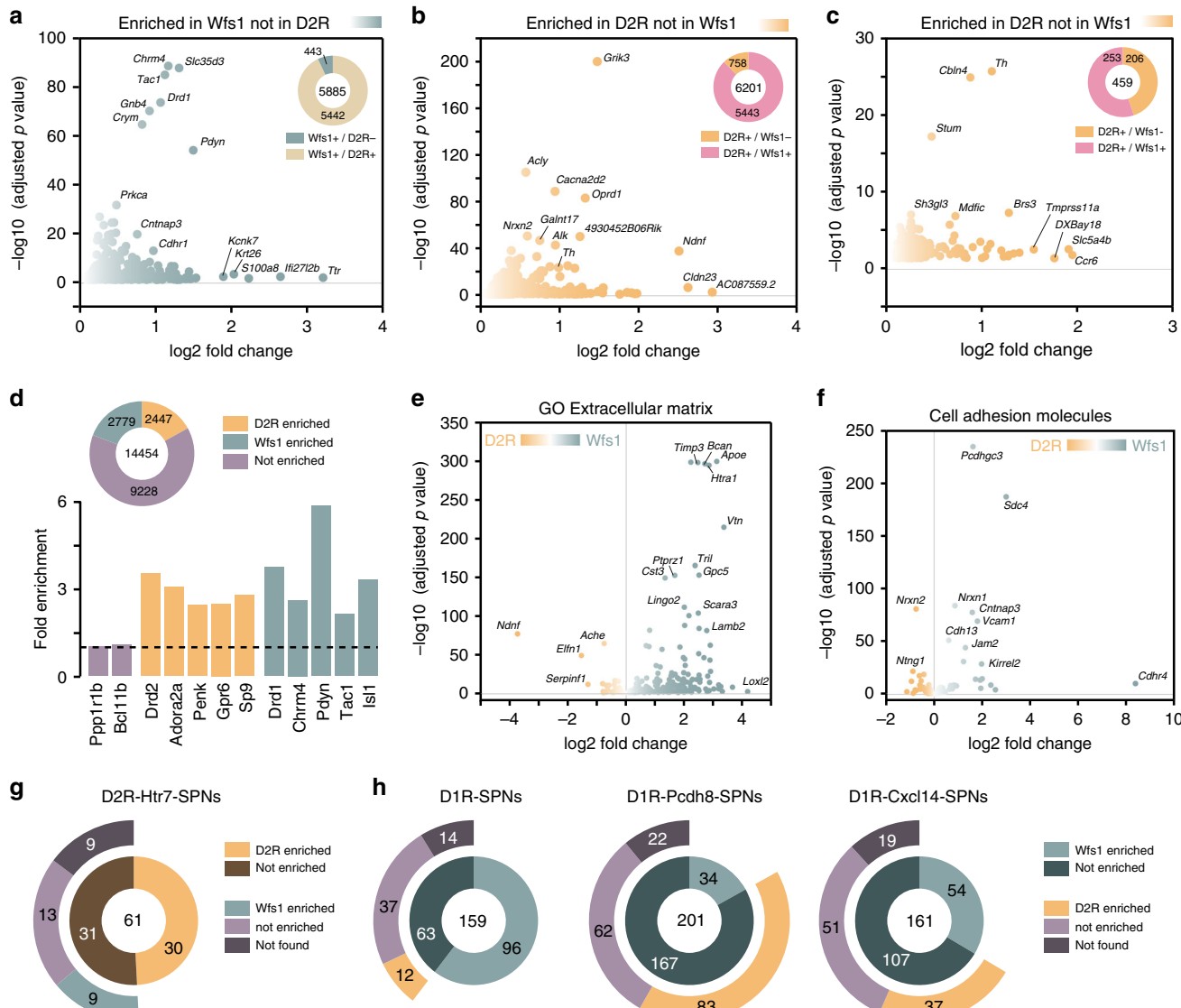

**Fig. 5 Molecular characterization of Acb WFS1 neurons. a** Volcano plot depicting protein-coding genes selectively enriched in the Acb of WFS1 neurons. Volcano plot depicting protein-coding genes selectively enriched in the Acb of D2R neurons without (**b**) or after filtering the pellet fraction with the input fraction (**c**) compared with Acb Wfs1 neurons. **d** Fold-change of markers enriched from the Acb pellet fraction of *Wfs1-RiboTag* mice vs *D2-RiboTag* mice. Volcano plots of GO enrichment analysis of genes related to extracellular matrix (**e**) including cell-adhesion molecules (**f**) in the Acb D2R and WFS1 neurons. (**g**) Doughnut chart showing the overlap and distribution of Acb D2R- and WFS1-enriched genes found in our study among the genes defining the D2R-Htr7 SPNs[13]. **h** Doughnut chart showing the overlap and distribution of Acb D2R- and WFS1-enriched genes found in our study among the genes defining the D1R-SPNs and some of their discrete subpopulations (D1R-Pcdh8-SPNs and D1R-Cxcl14-SPNs)[13].

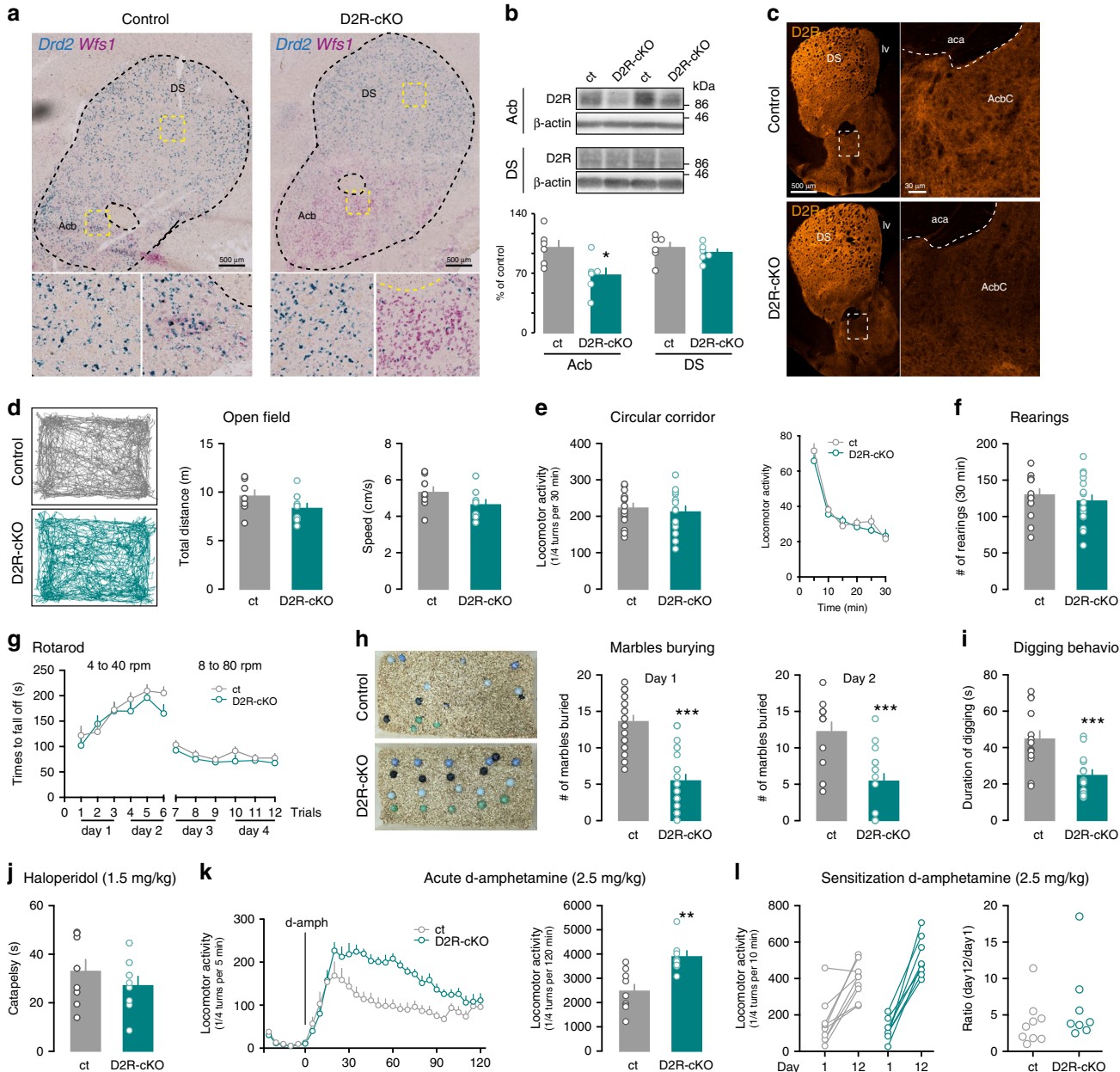

**Fig. 6 Temporal deletion of *Drd2* from WFS1 neurons alters digging behavior and amphetamine effects. a** Double ISH of *Drd2* (blue) and *Wfs1* (magenta) in the striatum of control and D2R-cKO mice. **b** WB (top) and quantification (bottom) of D2R in control (ct) and D2R-cKO in Acb ($t_{10} = 2.542$, $p = 0.0293$, two-sided *t* test, $n = 6$ mice/group) and DS ($t_{12} = 0.7706$, $p = 0.4588$, two-sided *t* test, $n = 6$ mice/group). **c** Coronal section of D2R staining in D2R-cKO and ct. **d** Representative track traces (left), total distance traveled over 30 min (middle) ($t_{13} = 1.51$, $p = 0.1549$, two-sided *t* test, $n = 7$ ct and $n = 8$ D2-cKO), and average speed (right) ($t_{14} = 1.563$, $p = 0.1405$, two-sided *t* test, $n = 8$ mice/genotype). Horizontal (**e**: $t_{28} = 0.8209$, $p = 0.4186$, two-sided *t* test, $n = 14$ ct and $n = 16$ D2R-cKO) and vertical (**f**: $t_{28} = 0.709$, $p = 0.4842$, two-sided *t* test, $n = 14$ ct and $n = 16$ D2R-cKO) activity over 30 min in a circular corridor. **g** Accelerating rotarod performance of D2R-cKO and ct. Time to fall off is represented among the six trials at 4–40 rpm (time: $F_{(5, 75)} = 16.47$, $p < 0.0001$; genotype: $F_{(1, 15)} = 0.7803$, $p = 0.3910$; interaction: $F_{(5, 75)} = 1.314$, $p = 0.2672$, two-way ANOVA repeated measures) and the six following trials at 8–80 rpm (time: $F_{(5, 75)} = 3.761$, $p = 0.0043$; genotype: $F_{(1, 15)} = 0.5911$, $p = 0.4539$; interaction: $F_{(5, 75)} = 0.7171$, $p = 0.6126$, two-way ANOVA repeated measures) (three trials/day for 4 days, $n = 16$ mice/genotype). **h** Representative pictures and number of marbles buried after 20 min for each genotype on day 1 ($t_{45} = 6.984$, $p = 1.07E{-}08$, two-sided *t* test, $n = 23$ ct and $n = 24$ D2R-cKO) and 24 h later (day 2) ($t_{29} = 4.154$, $p = 0.0003$, two-sided *t* test, $n = 15$ ct and $n = 16$ D2R-cKO). **i** Total duration of digging behavior over 3 min ($t_{27} = 3.913$, $p = 0.0006$, two-sided *t* test, $n = 14$ ct and $n = 15$ D2R-cKO). **j** Total catalepsy time 60 min after haloperidol administration (1.5 mg/kg) ($t_{14} = 0.9354$, $p = 0.3654$, two-sided *t* test, $n = 8$ mice/genotype). **k** Horizontal activity over 30 min of habituation and over 120 min after amphetamine administration (2.5 mg/kg) (time: $F_{(29, 435)} = 38.31$, $p < 0.0001$; genotype: $F_{(1, 15)} = 14.43$, $p = 0.0017$; interaction: $F_{(29, 435)} = 5.260$, $p < 0.0001$, two-way ANOVA repeated measures; $t_{15} = 3.799$, $p = 0.0017$, two-sided *t* test, $n = 9$ ct and $n = 8$ D2R-cKO) in a circular corridor. **l** Total locomotor activity over 10 min after amphetamine administration (2.5 mg/kg) on day 1 and after a challenge injection one week after a 5-day repeated treatment (day 12) and ratio day 12/day 1 ($t_{15} = 1.076$, $p = 0.2989$, two-sided *t* test, $n = 9$ ct and $n = 8$ D2R-cKO). All data are presented as mean values ± SEM. DS dorsal striatum, Acb accumbens, AcbC accumbens core, aca anterior commissure, lv lateral ventricle.

responses were observed in both genotypes after a single injection of the NMDA receptor antagonist, MK801 (0.3 mg/kg) (Supplementary Fig. 10j), suggesting that *Drd2* ablation from WFS1 subpopulation does not affect the glutamate contribution within the corticostriatal loop.

Together, these results show a role of D2R in WFS1 neurons in the acute locomotor effects induced by amphetamine but not in behavioral sensitization.

## Discussion

D2R play a crucial role in DA-mediated motor control and represents an important target to treat disorders (e.g., schizophrenia) in which DA signaling is altered. However, given the widespread expression of striatal D2R, the precise function of D2R in relation to their neuroanatomical locations within the striatum remains unclear. Here, based on our translatome analysis of D2R neurons, we have identified a specific group of striatal neurons in which *Drd2* ablation alters a specific type of behavior. Our results provide a proof of concept of the interest of identifying specific SPN subpopulations to better dissect the actions of DA on its target neurons.

The present work clearly supports the existence of the molecular heterogeneity of striatal D2R neurons. Importantly, our study reveals that this high level of diversity cannot be apprehended without taking into account the anatomical localization of those neurons throughout the dorso-ventral axis of the striatum. Indeed, our cross analysis indicated that expression patterns of enriched genes are highly heterogeneous within the DS, the AcbC, and the AcbSh. Although we focused here on the comparison of mRNAs of DS and Acb D2R neurons, similar expression patterns most likely exist for D1R-positive cells. Indeed, in a recent single-cell RNAseq study showing heterogeneity of SPNs subpopulations[13], neither whole-transcriptome PCA nor t-distributed stochastic neighbor embedding separated dSPNs from iSPNs within the neuronal cluster, suggesting that they share similar gene expression patterns. In line with this observation, we found that the biased dorso-ventral expression pattern of the *Wfs1* gene toward the Acb was found in both dSPNs and iSPNs.

Our data demonstrate the existence of a molecular complexity that goes beyond the classical D1R/D2R dichotomy[33,34], as iSPNs actually consist of several neuronal subpopulations with functional heterogeneity. In this regard, a recent study unveiled that D2R from the AcbSh have a higher sensitivity for DA than those from the DS. This difference has been attributed to postsynaptic signaling molecules that differ between the two regions[35]. In support of this hypothesis, our study revealed significant differences in expression of genes encoding for postsynaptic molecules that belong to G protein signaling and RGS family, which could account for the region-dependent D2R sensitivity to DA. Likewise, the present study further supports the existence of molecular heterogeneity among CINs. The biased expression of D2R toward DS CINs may explain why D2R-dependent hyperpolarization of CINs was preferentially observed in DS but not in AcbSh[32,36].

Using *Th-eGFP* mice, previous studies identified distinct classes of TH-positive cells in the striatum[37]. Despite the expression of VMAT1, TH neurons did not release DA and were clearly categorized as GABAergic interneurons[37,38]. However, our analysis revealed that a high proportion of TH cells co-expressed DARPP-32, indicating that at least a fraction of TH neurons within the Acb corresponds to D2R neurons. Our cross analysis strongly suggests that TH/DARPP-32 neurons located in the AcbSh may correspond to the recently identified D2R-Htr7-SPNs subpopulation[13], whose electrophysiological and morphological patterns as well as synaptic connectivity remain to be established.

Finally, our systematic gene classifications, GO analysis, and associated functional assays reveal distinct biological functions of D2R subpopulations in relation with their regional expression. Indeed, the predominant expression of mitochondria-related genes in D2R neurons from the DS is accompanied by an increased activity of CI. Such imbalanced mitochondrial activity and content could render DS iSPNs more vulnerable to oxidative stress in Huntington's disease, possibly explaining why they degenerate earlier than dSPNs and why the neuronal loss proceeds from dorsal to ventral striatum[39].

Our study unveils the role of DA in digging behavior via a mechanism that requires D2R in WFS1 neurons. Digging is an evolutionarily conserved trait used in many species to seek or hoard food, to create a refuge from cold or predators, or a nest for the young. These actions require both a motor component and a motivational value, two traits controlled by the Acb. In this line, our results point to a prominent role of D2R located in a subpopulation of the Acb in the control of digging behavior. Since Acb D2R neurons project to the VP, the Acb-VP subcircuit formed by WFS1 neurons is likely to be the one involved in DA-controlled digging. Importantly, decreased digging behavior is the only motor phenotype observed in D2R-cKO mice under basal conditions. In contrast to full D2R-KO mice[40,41] and lacking D2R in all iSPNs or in the Acb[5], no hypolocomotion or altered rotarod performance were observed in our D2R-cKO mice. Haloperidol-induced catalepsy was dampened in mice lacking the D2L isoform[30,31] or lacking D2R selectively from CINs[32] but remained unaltered in our D2R-cKO mice. Our study also reveals that boosting DA levels by the use of amphetamine induced a higher locomotor hyperactivity in D2R-cKO than in control littermates. This contrasts with the blunted response to cocaine—another psychostimulant that also increases DA levels—observed in SPNs D2R-KO mice[42], in mice lacking the D2L isoform[31], and in *Drd2^{loxP/loxP}* mice injected with a viral vector expressing Cre recombinase in the Acb[43]. The number of neurons bearing D2R ablation (DS and Acb) or possible developmental effects presumably account for these differences. Although the latter study targeted D2R deletion in the Acb during adulthood, the viral approach was not cell type-specific, hence leading to the knockdown of D2R in all D2R-expressing cells (D2R-SPNs, D1R/D2R-SPNs, and CINs) located within the Acb. In contrast, in our study D2R were ablated only from a specific iSPN subpopulation including a limited number of cells in the AcbC and intermediate AcbSh. Therefore, these opposing effects suggest that Acb D2R have different roles in response to drugs of abuse depending on the subcircuit in which they are located.

Collectively, here we identified novel region-specific and genetically defined D2R cell subpopulations, which will likely be useful for delineating and studying discrete striatal subcircuitry. Manipulation of D2R from one subcircuit, WFS1-positive neurons, produces a highly specific behavioral phenotype, and suggests a selective role for D2R in these neurons in the control of digging behavior and amphetamine-induced hyperlocomotion. Therefore, our approach opens the way for parsing the behavioral role of D2R in specific, well-defined groups of striatal neurons.

## Materials and methods

**Animals**. The different mouse lines used in the present study are listed in the resource table (Supplementary Information). Homozygous *RiboTag* female mice were crossed with *D2R-Cre* heterozygous male and *Wfs1-CreERT2* heterozygous male mice to generate *D2-RiboTag* and *Wfs1-RiboTag*, respectively. To delete *Drd2* from WFS1 neurons, heterozygous *Wfs1-CreERT2* mice were crossed with homozygous *Drd2^{loxP/loxP}* mice. First-generation animals expressing Cre under Wfs1 promoter were crossed a second time with homozygous *Drd2^{loxP/loxP}* mice to generate the tamoxifen-inducible *Drd2* ablation specifically in WFS1 neurons (D2R-cKO). For all behavioral experiments, male and female homozygous *Drd2^{loxP/loxP}* mice expressing CreERT2 under *Wfs1* regulatory sequence were

compared with controls (Cre-negative). Mice were housed under standardized conditions with a 12 h light/dark cycle, stable temperature (22 ± 2 °C), controlled humidity (55 ± 10%), and food and water ad libitum. Housing and experimental procedures were approved by the French Agriculture and Forestry Ministry (A34-172-13). Experiments were performed in accordance with the animal welfare guidelines 2010/63/EC of the European Communities Council Directive regarding the care and use of animals for experimental procedures.

**Drugs and treatments**. (5R,10S)-(+)-5-Methyl-10,11-dihydro-5H-dibenzo[a,d]cyclohepten-5,10-imine hydrogen maleate ((+)-MK801, 0.3 mg/kg) and haloperidol (0.5 mg/kg) were purchased from Tocris Bioscience, and D-amphetamine hemisulfate salt (2.5 mg/kg), clozapine N-oxide (CNO, 1 mg/kg), and tamoxifen (100 mg/kg) from Sigma-Aldrich. All drugs were administered intraperitoneally in a volume of 10 ml/kg and dissolved in 0.9% (w/v) NaCl (saline), except CNO that was dissolved in 0.1% DMSO and tamoxifen that was dissolved in sunflower oil/ethanol (10:1) to a final concentration of 10 mg/ml.

**Tissue collection**. Mice were killed by cervical dislocation and the heads were immersed in liquid nitrogen for 4 s. The brains were then removed and sectioned on an aluminum block on ice. The whole striatum was extracted as previously described[44]. The Acb was isolated from a ~1-mm thick coronal section located between 1.94 and 0.86 mm anterior to bregma and the DS between 0.86 and 0.14 mm anterior to bregma as previously described[45] (Supplementary Fig. 1a).

**Western blot**. The DS and the Acb were sonicated in 200 μl of 10% sodium dodecyl sulfate and boiled at 100 °C for 10 min. Protein quantification and WBs were performed as described[45]. The antibodies used are summarized in a Supplementary resource table. When necessary, membranes were stripped in buffer containing 100 mM glycine (pH 2.5), 200 mM NaCl, 0.1% Tween 20, and β-mercaptoethanol for 45 min, followed by extensive washing in 100 mM NaCl, 10 mM Tris, and 0.1% Tween 20 (pH 7.4). For quantitative purposes, the optical density values of each antibody were normalized to β-actin values from the same sample. Acb values were referred as 100% as control group and DS values were represented with respect to Acb. In all blots, quantified data are shown as the mean ± SEM ($n = 7$ mice/brain subregion) (Supplementary Fig. 12).

**Immunofluorescence**. Tissue preparation and IF were performed as described[45]. Primary antibodies used are listed in the resource table (Supplementary Information). Three slices per mouse were used in all IF analyses ($n = 3$–4 mice/staining).

**In situ hybridization**. Staining for Wfs1 and Drd2 mRNAs was performed using single molecule fluorescent ISH (smFISH). Brains from two C57Bl/6J 8-week-old male mice were rapidly extracted and snap-frozen on dry ice and stored at −80 °C until use. Ventral striatum coronal sections (14 μm) were collected directly onto Superfrost Plus slides (Fisherbrand). RNAscope Fluorescent Multiplex labeling kit (ACDBio Cat No. 320850) was used to perform the smFISH assay according to the manufacturer's recommendations. Probes used for staining are mm-Wfs1-C2 (ACDBio Cat No. 500871-C2) and mm-Drd2-C3 (ACDBio Cat No. 406501-C3). After incubation with fluorescent-labeled probes, slides were counterstained with DAPI and mounted with ProLong Diamond Antifade mounting medium (Thermo Fisher Scientific, P36961). Fluorescent images were captured using sequential laser scanning confocal microscopy (Leica SP8). Basescope Duplex Assay was used for the simultaneous visualization of Wfs1 mRNA and the selective detection of exon 2 of Drd2 mRNA. BaseScope assay was performed and assisted following guidelines (BaseScope™ Detection Reagent Kit-RED User Manual) provided by the supplier (ACDBio Cat No. 323810). Probes used for staining are BA-Mm-WFS1-4EJ-C2 (ACDBio Cat No. 724201-C2) and BA-Mm-Drd2-4zz-st1 (ACDBio custom made No. 724211). Slides were counterstained with hematoxylin and images were captured using brightfield microscope.

**Polyribosome immunoprecipitation**. HA-tagged-ribosome immunoprecipitation was performed as described previously[46] in the whole striatum, DS, and Acb of D2-RiboTag mice and in Acb of Wfs1-RiboTag mice. Total RNA was extracted from ribosome-mRNA complexes using RNeasy Microkit (Qiagen) followed by in-column DNAse treatment to remove genomic DNA contamination. Quality and quantity of RNA samples were both assessed using Agilent Bioanalyzer 2100 (Agilent Technologies). Three biological replicates, each one composed of a pool of 3–4 mice, were used for RNAseq analysis.

**cDNA synthesis and quantitative real-time PCR**. After HA-tagged-ribosome immunoprecipitation in a different cohort of D2-RiboTag mice than those used for RNAseq, synthesis of cDNA and qRT-PCR were performed as previously described[46] ($n = 5$ mice/brain subregion). Results are presented as linearized Cp-values normalized to housekeeping genes β-actin or Hprt2 and the ΔΔCP method was used to give the fold-change. Primer sequences are indicated in the resource table (Supplementary Information).

**Stranded mRNA library preparation and sequencing**. The libraries from the mouse total RNA were prepared using the TruSeq® Stranded mRNA LT Sample Prep Kit (Illumina, Inc., Rev.E, October 2013) according to the manufacturer's protocol. Briefly, 0.25 μg of total RNA was used for poly-A-based mRNA enrichment with oligo-dT magnetic beads. The mRNA was fragmented (resulting RNA fragment size was 80–250 nt, with the major peak at 130 nt) and the first strand cDNA synthesis was done by random hexamers and reverse transcriptase. The second strand cDNA synthesis was performed in the presence of dUTP instead of dTTP, this allowed to achieve the strand specificity. The blunt-ended double stranded cDNA was 3′ adenylated and Illumina platform compatible adaptors with unique dual indexes and unique molecular identifiers (Integrated DNA Technologies) were ligated. The ligation product was enriched with 15 PCR cycles and the final library was validated on an Agilent 2100 Bioanalyzer with the DNA 7500 assay. Each library was sequenced on HiSeq4000 (Illumina) in a fraction of a HiSeq 4000 PE Cluster kit sequencing flow cell lane, following the manufacturer's protocol for dual indexing. Image analysis, base calling and quality scoring of the run were processed using the manufacturer's software Real Time Analysis (RTA 2.7.7) and followed by generation of FASTQ sequence files.

**Bioinformatic analysis**. RNAseq reads were mapped against the mouse reference genome (GRCm38) using STAR version 2.5.3a[47] with ENCODE parameters for long RNA. Annotated genes (gencode version M21) were quantified using RSEM version 1.3.0 with default parameters[48]. PCA was done using the top 500 most variable genes with the "prcomp" R function and "ggplot2" R library after the regularized log (rlog) transformation of the counts. Differential expression analysis was performed with DESeq2 version 1.18.1[49] with a prefiltering on lowly expressed genes (at least ten normalized reads for each sample in one group). Differentially expressed genes were considered those with FDR < 0.05 and absolute shrunken fold-change > 1.5. Heatmaps with the top 50 differentially expressed genes were performed with the pheatmap R package[50] with the scaled rlog counts. Functional enrichment analysis of the differentially expressed genes was performed with Gprofiler[51].

**Mitochondria assays**. DS and Acb were dissected and stored at −80 °C until further use. Tissue ($n = 20$–22/brain area) was homogenized and processed as previously described[52] in order to measure CI (EC 1.6.5.3) and CS (E.C. 2.3.3.1). Assay measurements were performed in duplicate.

**Stereotaxic injections**. Surgeries were performed on 7-week-old Wfs1-CreERT2 mice. Mice were anesthetized with a mixture of ketamine (Imalgene 500, 50 mg/ml, Merial), 0.9% NaCl solution (weight/vol), and xylazine (Rompun 2%, 20 mg/ml, Bayer) (2:2:1, i.p., 0.15 ml/30 g) and mounted on a stereotaxic apparatus. The microinjection needle was connected to a 10 μl Hamilton syringe and filled with adeno-associated virus (AAV) containing Gs-DREADD (pAAV-hSyn-DIO-rM3D (Gs)-mCherry) for tracing studies and Gq-DREADD (pAAV-hSyn-DIO-hM3D (Gq)-mCherry) for chemogenetic studies (Penn Vector Core facility). A total volume of 0.5 μl was injected bilaterally into the Acb (A/P = 1.54 mm; M/L = ±1.3 mm; DV = −4.7 mm from bregma) over 10 min and the needle was left in place for an additional 5 min to allow for diffusion of viral particles away from injection site. Mice were allowed to recover for 2 weeks, then treated for 3 days with tamoxifen (100 mg/kg), and either perfused 3 weeks later for tracing studies or treated with clozapine N-oxide (CNO) (1 mg/kg) and tested 30 min later for chemogenetic studies.

**Behavioral assays**. D2R-cKO mice and control littermates (7–8 weeks old) were all treated 3 weeks prior to testing with tamoxifen (100 mg/kg) for 3 consecutive days. Wfs1-CreERT2 mice (8–16 weeks old) were treated with vehicle or CNO 30 min before testing. Mice were also handled for 3 days prior to testing for habituation. Since there was no evidence of sex differences in our behavioral measurements, data from male and female mice were pooled. All experiments were blinded to genotype during behavioral testing.

**Locomotor activity**. Spontaneous exploratory behavior was measured in an open field (white plastic box, 35 cm width × 45 cm length × 25 cm height) for 30 min. Videos were analyzed using Noldus Ethovision software, and total distance traveled and average speed were calculated. Horizontal and vertical activity was measured in a circular corridor (Imetronic, Pessac, France) for 30 min. Counts for horizontal activity were incremented by consecutive interruption of two adjacent beams placed at a height of 1 cm per 90° sector of the corridor (mice moving through 1/4 of the circular corridor) and counts for vertical activity (rearings) corresponding to interruption of beams placed at a height of 7.5 cm along the corridor (mice stretching upward) were used as an additional measure for exploratory activity. For amphetamine administration, mice were first habituated for 3 consecutive days in the same circular corridor used to measure locomotor activity. In this habituation phase, mice were placed in the activity box for 30 min, received a saline injection, and returned in the box for 2 h. On day 4, mice underwent the same procedure except that amphetamine (2.5 mg/kg) was administered instead of saline. For sensitization studies, mice received amphetamine repeatedly for 5 consecutive days, followed a 7-day withdrawal period, and on day 12 received a challenge injection of

amphetamine (2.5 mg/kg). Locomotor activity was measured as on day 4. For MK801 administration, mice were habituated for 30 min before MK801 treatment (0.3 mg/kg).

**Marble burying test**. Marble burying test was performed as previously described[53] except the duration of the test that was 20 min.

**Digging test**. Digging test was performed as previously described[53]. Before goal-directed digging experiments mice were habituated to palatable food to avoid neophobia. Mice were food-deprived the day before the test.

**Grooming test**. Grooming test was performed as previously described[54] except for the duration that was 10 min.

**Novel object exploration**. Novel object exploration test was performed as previously described[5].

**Accelerating rotarod**. Rotarod testing was performed over the course of 4 days as previously described[29].

**Elevated plus maze**. Elevated plus maze was performed as previously described[55].

**Catalepsy**. Mice were injected with haloperidol (1.5 mg/kg), dissolved in 0.2% acetic acid, and returned to their home cage until testing. Each animal was individually positioned 60 min post injection so that its hindlimbs were on a plane surface and its forelimbs rested on a 0.3 cm diameter horizontal bar, 6 cm above the surface level. Once the mouse remained immobile after release, the time until the first removal of a front paw or sustained head movement was recorded by stopwatch to a maximum of 180 s.

**Tail suspension test**. Mice were suspended 50 cm above a cushioned pad using tape to attach their tails to a horizontal pole above the pad. Each mouse was tested for a 6 min trial. Latency to the first bout of immobility (defined as ≥5-s-long segment of time spent immobile) and the total time spent immobile during the trial were recorded. Immobility was defined as hanging passively without any movement of the head or paws.

**Spontaneous alternation test**. Spontaneous alternation was measured in a Y-shaped maze with three identical arms ($40 \times 9 \times 16$ cm) at a 120° angle. Each individual mouse was placed in the center of the maze and allowed to freely explore for 5 min. A triad was defined as a set of three arm entries, when each entry was to a different arm of the maze. The number of arm entries and the number of triads were recorded. The percentage of alternation was calculated by dividing the number of triads by the number of possible alternations and then multiplying by 100.

**Three-chamber social approach**. A three-chamber arena was used to assess sociability and preference for social novelty. On day 1, stranger target mice were habituated to the wire cups. On day 2, test mice (2-month-old C57BL/6 males or females) were placed in the middle chamber and allowed to freely explore all the empty chambers of the apparatus for 10 min. Next, an unfamiliar mouse (Stranger#1, gender matched) was introduced into one of the two side chambers, enclosed in a wire cage allowing only for the test mouse to initiate any social interaction. An identical empty wire cage was placed in the other side chamber. Following placement, the test mouse was allowed to explore the whole three-chamber arena for 10 min. At the end of the 10 min sociability test, a new unfamiliar mouse (Stranger#2, gender matched) was placed in the previously unoccupied wire cage, and test mice were examined for an additional 10 min to assess preference for social novelty. The time spent sniffing the Stranger#1, Stranger#2, or empty wire cages were manually scored. The discrimination index for sociability and social novelty were calculated as follows, respectively: (time exploring Stranger#1—time exploring empty wire cage)/(total exploration time) × 100 and (time exploring Stranger#2—time exploring Stranger#1)/(total exploration time) × 100.

**Statistical analyses**. GraphPad Prism v6.0 software was used for statistical analyses. Data are shown as the means ± SEM. For normally distributed parameters, Student's $t$ test (unpaired, two-sided) was used for all the tests except the rotarod, which was analyzed by two-way ANOVA repeated measures. $*p < 0.05$, $**p < 0.01$, and $***p < 0.001$.

**Reporting summary**. Further information on research design is available in the Nature Research Reporting Summary linked to this article.

## Data availability
Sequence data have been deposited in Gene Expression Omnibus, accession code GSE94145. The data supporting the findings of this study are available within the paper and its Supplementary materials files or available from the corresponding author upon reasonable request. The IUPHAR/BPS database (www.guidetopharmacology.org) was used to implement gene classification.

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

## Acknowledgements

We thank Gabriel Mel de Fontenay for critical reading of the manuscript and Denis Hervé and Pierre-François Mery for their insightful comments. We thank Elisabeth Poole for the anti-Peg10 antibody. We thank iExplore and MRI Platforms. This work was supported by Inserm, Fondation pour la Recherche Médicale (DEQ20160334919) (E.V.) and (DPA20140629798) (J-A.G. and E.V.), La Marato de TV3 Fundacio (#113-2016) (E.V. and M.M.), and Agence National de la Recherche (EPITRACES, ANR-16-CE16-0018) (E.V. and J-A.G.), NARSAD Young Investigator Grant from the Brain and Behavior Research Foundation (E.P.), NS091144 and NS103037 from NIH/NINDS (J.B.D.), European Research Council (ERC-2014-StG-638106), MINECO Ramon y Cajal fellowship (RyC-2012-11873) and AGAUR (2017SGR-323) (A.Q.), and MICIU Proyectos I+D RETOS-JIN (RTI2018-101838-J-I00) (E.S.). A.E.-C. is funded by ISCIII/MINECO (PT17/0009/0019), which is co-funded by FEDER. E.P. was a recipient of Marie Curie Intra-European Fellowship IEF327648 and Visiting Scholar Junior Fellowship (France-Stanford Center for Interdisciplinary Studies). E.P. is currently a recipient of Beatriu de Pinós fellowship (#2017BP00132) from University and Research Grants Management Agency (Government of Catalonia). L.C. is supported by the Labex Epi-GenMed («Investissements d'avenir», ANR-10-LABX-12-01).

## Author contributions

E.P. and E.V. conceived and led the project. E.P., E.V., and M.M. designed the study. E.P., E.V., and J.B.D. conceptualized the project. E.P. and M.M. performed brain dissections. E.P. performed biochemical experiments. E.P. and L.C. performed histological and behavioral experiments. S.M. and G.M. performed mitochondrial assays. E.P., C.Z., and K.K. performed stereotaxic injections. E.P. and J.B-V. performed qRT-PCR. A.E.-C. and S.R. performed bioinformatic analyses. M.G. led RNAseq. C.K., M.R., M.M., and J-A.G. provided mice. E.S., A.Q., J.B.D., and J-A.G. provided reagents. E.V. performed gene classifications and supervised the project. E.P. and E.V. wrote the manuscript with input from all authors.

## Competing interests

The authors declare no competing interests.
