## [Peer Review File · Nature Communications]

Reviewers' Comments:

Reviewer #1:

Remarks to the Author:

In this manuscript, Puighermanal et al. employ a broad range of experimental approaches in efforts to define the heterogeneity of D2R-expressing cells along the dorso-ventral striatal axis. The concept of exploring striatal diversity as a function of anatomical location is an interesting and likely productive direction. However, this paper, by not focusing on one specific issue, does not provide sufficient novel insight to currently warrant publication in Nature Communications. The authors have two interesting stories that are both partially developed in parallel but do not synergize in any way to improve our understanding. This work would benefit from deeper study of the fascinating behavioral phenotype. The transcriptional profiling is thorough and interesting and might better stand alone as a resource-type publication.

Major Issues:

1. the Wfs1+ striatal sub-circuit. This is an interesting concept that is not sufficiently developed. The authors do a solid job describing how this population relates to D1/D2 expression but don't offer much more to detail the specificity of this "subcircuit." What makes this Wfs1+ population unique beyond their Wfs1+ expression? Is there some underlying molecular, anatomical or physiological difference? There is some general anatomy – however, are there specific A-P,D-V locations where the Wfs1+ population targets? Might analysis of this population benefit from a comparison with the NAc Wfs1- population (accessible by CRE-OFF viral tools, Saunders, Current Protocol 2015)? This seems like a perfect place to synergize the transcriptional data with the behavior by specifically looking at the Wfs1+ transcriptome and comparing it to NAc Wfs1- populations. Another way to demonstrate potential specificity in a circuit would be to use the Wfs1-CreER line to gain optogenetic control and explore how activation/inhibition of this population might affect behavior.
2. the authors propose a very specific role for D2Rs in Wfs1+ circuits – the control of digging. There is much to be further explored in this behavior – the data suggests that the duration of individual digging bouts is reduced. Does this mean that D2Rs in these neurons are important for digging or just the continuation of ongoing motor behaviors? Are the kinematics of the digging bouts normal? If you buried food (goal-directed digging), would the same abnormalities manifest? As mentioned above, here is another instance where optogenetic activation of the Wfs1+ population may help to specifically determine whether the circuit specifically drives digging or other has a role in continuing a range of motor states.
3. the pharmacological experiments provide superficial understanding of the role of Wfs1+ D2R+ SPNs. For example, the haloperidol experiment: is the implication that the haloperidol-sensitive population dorsal striatal D2Rs or the Wfs1- population of NAc D2R+ neurons? For the systemic MK-801: how does this specifically relate to striatal afferents – how is this specificity achieved?
4. the logic for doing a D2R transcriptome is not totally clear given that there are other local striatal interneurons that will contaminate this data set. As the authors state (and then show) ChAT interneurons differentially contaminate the D vs V samples. It would seem very hard to specifically attribute transcripts at the population level to either cell class under these conditions. What about the use of the A2a-Cre line instead to only label D2R+ SPNs?
5. for the transcriptional profiling, the authors perform an extensive "functional" analysis of D2R-expressing cells in the D and V striatum. However it seems excessive to claim "that the DS- and Acb-enriched genes have distinct biological functions" based solely on GO-type analyses. Furthermore, there is an even larger leap from this to "these results indicate that D2R neurons have distinct biological functions depending on their anatomical localization within the striatum...."

Minor Points:

1. description of D2R LOF might be better understood via ISH as opposed to immuno. As the authors themselves report, the presence of D2R protein in cortical terminals, ChAT interneurons makes the immuno hard to interpret. ISH would show the true percentage of Wfs1+ neurons that have undergone recombination.
2. what is this non-striatal expression of Wfs1+ cells in cortex? Do they express D2Rs or can they be ignored?
3. line 71, widespread or non-specific instead of "miscellaneous"
4. "no expression of Wfs1 was detected in CINs..." This is important and should be shown.
5. perhaps specific phases of grooming could be analyzed, to see whether there are changes in the duration of discrete motor states – another way to test whether the digging phenotype is really specific to digging.

Reviewer #2:

Remarks to the Author:

This manuscript addresses two questions. By using a Wfs1-CreERT in a floxed D2R mouse the authors were able to knockout D2R in a subset of cells expressing this receptor primarily in the Acb region of the ventral striatum. These partial KO mice displayed a phenotype characterized by decrease digging and marble burying as well as increased locomotor activity in response to amphetamine. A separate question concerned the transcript profiles of D2R neurons in the ventral versus dorsal striatum and this was addressed by using a ribosome tagging approach driven by a D2R-Cre crossed with Ribotag.

Critique:

1. These two questions are not really related and I question whether they belong in the same paper. It would have been interesting to determine the transcripts expressed in the subset of D2R neurons that express Wfs1 compared to those that do not but this would require more sophisticated analysis.
2. The projections from the Wfs1 neurons were partially traced but the actual circuit downstream of the Wfs1/D2R neurons was not determined. The behavioral changes with this partial KO are interesting but the authors need to follow up with a more detailed mechanistic study to make it meaningful.
3. The D2R-Cre/Ribotag study is not properly controlled. It appears that all of the RNA-seq was done on the precipitated polysome fraction without comparison to the inputs for the Acb or dorsal striatum (DS). In the QPCR experiment shown in Fig. 3, the enrichment over total RNA in the D2R neurons appears to be 2-4 fold and the de-enrichment of D1R is about 4 fold. There is significant background expected for these IP experiments and the only way to filter this out is to determine which genes are significantly enriched in the IP fraction. This would require analyzing the inputs for each experiment by RNA-seq and then filtering out the genes which are not expressed or enriched in D2R neurons
4. The D2R-Cre was not conditional so any cell that expressed D2R at any time during development would activate the Ribotag in all of the differentiated cells that develop from those progenitors. Is there any literature on the developmental expression of D2R in the embryo?
5. Given the detailed single cell analysis of mRNA expression in the striatum by Gokee et al in 2016 it is unclear how this manuscript adds significantly to that information. The Gokee group did not do an anatomical selection for their study but the Allen Brain ISH allows one to identify where marker genes coexpressed with D2R are localized as the authors show in supplemental Fig3 of this manuscript.

Reviewer #3:

Remarks to the Author:

In their manuscript, Puighermanal et al. describe a collection of experiments related to D2R-expressing neurons in the dorsal striatum/accumbens. There are a number of results, which I will try to summarize here:

1. When D2R expression was reduced in *wfs1* neurons (found mostly in the accumbens), mice exhibit specific behavioral deficits in digging and marble burying, but not in any other of a number of behavioral tasks. These same mice also exhibited exaggerated amphetamine-induced locomotor responses.
2. The authors used D2R-Cre:RiboTag mice to isolate ribosome-bound RNAs from D2R-expressing neurons in either the dorsal striatum or the accumbens. A number of genes were found that were preferentially expressed in one compartment or the other which were then validated with qRT-PCR and in some cases, with immunostaining.
3. Using available data bases, the authors then classified genes identified as exhibiting a compartment-specific gradient.
4. Finally, the authors performed gene ontology analysis to further analyze their data, but I found this section hard to follow.

Overall, this manuscript seemed like a series of observations, not necessarily related, combined into a single manuscript. The behavioral results in particular seemed quite disconnected from the rest of the manuscript, and I would suggest that the behavior and gene expression work be separated in any subsequent manuscripts. The behavioral results are interesting, but preliminary.

Responses to Referees

We are thankful for the referees' evaluation of our manuscript and for their positive comments and constructive feedback. We are delighted that our work was described as "*the concept of exploring striatal diversity as a function of anatomical location is an interesting and likely productive direction*", "*The transcriptional profiling is thorough and interesting*".

The reviewers raised important questions about our data and provided valuable suggestions. We have now made several arrangements following the suggestions of the reviewers. We believe this new revised version of the manuscript is substantially improved. Specific responses to the reviewers' comments are provided below and the main modifications have been highlighted in yellow in the revised version of the manuscript.

Major modifications

In order to address reviewers' requests several reorganizations have been made in the manuscript.

1) As requested by referee #2 we have now performed a new high-throughput RNAseq including the comparison between pellets and inputs in both DS and Acb. These new experiments are now presented in the **Figure 1, Figure S3** and **Table S1, S2** and **S3**. Our results have been now also analyzed in light of the recent single-cell RNAseq studies (Gokce et al., 2016 and Saunders et al., 2018).

2) Because all the reviewers raised similar concerns about the framing of the paper, we have now reorganized the manuscript. Moreover, a complete molecular characterization of the Acb Wfs1 neurons is now provided in the revised version. These new results, presented in **Figure 4** and **Figure 5**, allowed us to compare the transcriptome of Acb D2R and Wfs1 neurons.

We provide a point-by-point discussion of the referee's comments below, and hope they find our revised study suitable for publication in *Nature Communications*.

Reviewer #1:

In this manuscript, Puighermanal et al. employ a broad range of experimental approaches in efforts to define the heterogeneity of D2R-expressing cells along the dorso-ventral striatal axis. The concept of exploring striatal diversity as a function of anatomical location is an interesting and likely productive direction. However, this paper, by not focusing on one specific issue, does not provide sufficient novel insight to currently warrant publication in Nature Communications. The authors have two interesting stories that are both partially developed in parallel but do not synergize in any way to improve our understanding. This work would benefit from deeper study of the fascinating behavioral phenotype. The transcriptional profiling is thorough and interesting and might better stand alone as a resource-type publication.

We thank the reviewer for his/her positive comments. We have now reorganized the manuscript and performed several additional experiments to link what was perceived as two stories.

Major Issues:

- the Wfs1+ striatal sub-circuit. This is an interesting concept that is not sufficiently developed. The authors do a solid job describing how this population relates to D1/D2 expression but don't offer much more to detail the specificity of this "subcircuit." What makes this Wfs1+ population unique beyond their Wfs1+ expression? Is there some underlying molecular, anatomical or physiological difference? There is some general anatomy – however, are there specific A-P,D-V locations where the Wfs1+ population targets? Might analysis of this population benefit from a comparison with the NAc Wfs1- population (accessible by CRE-OFF viral tools, Saunders, Current Protocol 2015)? This seems like a perfect place to synergize the transcriptional data with the behavior by specifically looking at the Wfs1+ transcriptome and comparing it to NAc Wfs1- populations

To further characterize molecular identity of the Acb Wfs1 population we have generated *Wfs1-Ribotag* mice and performed RNAseq of the tagged ribosome-bound mRNAs and the input fractions of the Acb. Thus, we elucidated –for the first time– the translome profile of this novel striatal subpopulation. As the reviewer pointed out, beyond Wfs1+ expression, we identified additional 5887 genes that are enriched in Wfs1-expressing neurons. Additionally, these samples have been sequenced at the same time than the others (tagged ribosome-bound mRNAs and the input fractions of the Acb and DS of the *D2-RiboTag* mice) allowing us to perform a direct comparison of the translome of Acb D2R vs Wfs1 neurons. These new results are now presented in **Figures 4** and **5**. Last, since we generated new samples from *D2-RiboTag* mice to sequence them again at the same time than those from *Wfs1-RiboTag* mice, we have updated all the RNAseq graphs as well as gene classifications in **Figures 1, 2, 3, S5, S6, S7, S8, and S9**.

- Another way to demonstrate potential specificity in a circuit would be to use the Wfs1-CreER line to gain optogenetic control and explore how activation/inhibition of this population might affect behavior

To further examine the circuit of Acb Wfs1-positive neurons in digging behavior we also assessed the optogenetic experiment suggested by the reviewer. We bilaterally injected a Cre-dependent adeno-associated viral vector expressing channelrhodopsin-2 (ChR2) in the Acb of Wfs1-CreERT2 mice. Four weeks later, we bilaterally implanted optic fibers either in the ventral pallidum or in the substantia nigra reticulata. One week later, we photostimulated the terminals of Wfs1 neurons and measured both locomotor activity and digging behavior. Even though mice were plugged to the patch cord and habituated to the arena for 1 h during the 3 days prior to the test, unfortunately, it seemed that the fiber optic cable that transmits the light from the laser diode to the implanted optic fiber cannula in the mouse brain was too heavy and interfered with the spontaneous mouse behavior. Thus, all mice showed a decreased digging behavior when they were plugged to the optic cord independently if the laser was ON or OFF. In Figure 1 (below), the low digging duration and number of digging bouts of mice used in the optogenetic study is shown in comparison to the measurements obtained in naïve mice that were not plugged to the cable (from the same strain, C57BL/6). Given that the digging values may

likely be at a bottom effect, we could not observe an effect of the stimulation of Wfs1+ neurons terminals.

Figure 1. Optogenetic experiment. Digging behavior (upper panel) and locomotor activity (lower panel) measurements in Wfs1-CreERT2 mice injected with AAV-expressing ChR2 in the Acb and implanted with optic fibers either in the ventral pallidum (VP) or in the substantia nigra reticulata (SNr). Blue light stimulation was ON/OFF every 3-min intervals during 30 min. Behavior measurements are represented as the sum of the five 3-min blocks when the light was ON or OFF.

In addition to the optogenetic approach, we also performed chemogenetic experiments to further explore how the activation of Wfs1 neurons affects behavior. To do so, we bilaterally injected a Cre-dependent adeno-associated viral vector expressing Gq-coupled hM3D-mCherry in the Acb of Wfs1-CreERT2 mice. Four weeks later we tested a battery of behavioral tests 30 min after clozapine-N-oxide (CNO) or saline administration. No effect was found in locomotion, anxiety, novel object exploration, sociability, perseverance, spontaneous alternation, grooming, or digging behavior. In our study, we found that D2R from Acb Wfs1 neurons play a key role in digging behavior, whereas in our chemogenetic experiment, we did not find any effect in this particular behavior (Figure 2, below). This difference could be due to the fact that Wfs1 is expressed in both dSPNs and iSPNs and by activating neurons of the direct and indirect pathway at the same time, the effect is compensated. We decided not to include this data in the manuscript since it goes beyond the scope of addressing the region-specific D2R's role on behavior.

Figure 2. Chemogenetic experiment. Total digging duration over 3 min (a), latency to start digging (b), and number of digging bouts over 3 min (c) in Wf1-CreERT2 30 min after saline or CNO administration (1 mg/kg, ip)

- the authors propose a very specific role for D2Rs in Wfs1+ circuits – the control of digging. There is much to be further explored in this behavior – the data suggests that the duration of individual digging bouts is reduced. Does this mean that D2Rs in these neurons are important for digging or just the continuation of ongoing motor behaviors? Are the kinematics of the digging bouts normal?

When assessing locomotor activity or grooming, no alteration in the continuation of the ongoing behavior was observed. It is therefore likely a selective effect on digging.

- If you buried food (goal-directed digging), would the same abnormalities manifest? As mentioned above, here is another instance where optogenetic activation of the Wfs1+ population may help to specifically determine whether the circuit specifically drives digging or other has a role in continuing a range of motor states.

We have now performed additional experiments suggesting that goal-directed digging is not affected (**Figure S10**). Because Wfs1 is expressed in both D1R and D2R cells, future intersectional approaches need to be developed to specifically determine the functional role of Wfs1/D2R- vs Wfs1/D1R-populations. We believe this important question, which requires tools not available such as Drd2-flp mice, goes beyond the scope of the present study.

- the pharmacological experiments provide superficial understanding of the role of Wfs1+ D2R+ SPNs. For example, the haloperidol experiment: is the implication that the haloperidol-sensitive population dorsal striatal D2Rs or the Wfs1- population of NAc D2R+ neurons? For the systemic MK-801: how does this specifically relate to striatal afferents – how is this specificity achieved?

Both haloperidol-induced catalepsy and MK801-induced hyperactivity require D2R expressed in SPNs (Kharkwal et al., 2016). Our results clearly demonstrate that temporally-controlled inactivation of D2R in a discrete population of Acb SPN does not affect these behavioral responses. These results further emphasize the need to study the role of D2R in specific striatal subcircuits.

- the logic for doing a D2R transcriptome is not totally clear given that there are other local striatal interneurons that will contaminate this data set. As the authors state (and then show) ChAT interneurons differentially contaminate the D vs V samples. It would seem very hard to specifically attribute transcripts at the population level to either cell class under these conditions. What about the use of the A2a-Cre line instead to only label D2R+ SPNs?

One of the aims of this study was to identify striatal region-specific markers that could serve as tools to manipulate D2R from novel genetically-defined cell subpopulations. Therefore, we provide a comprehensive description of the transcriptome of DS vs Acb D2R neurons which include both iSPNs and CINs. Based on the number of reads and cross-analysis with anatomical and recent single-cell RNAseq studies we were able to clearly identify genes enriched in iSPNs (the vast majority) and CINs. Moreover, our analysis also provided important new insights regarding the molecular heterogeneity among CINs, with a biased expression of D2R along the dorso-ventral axis. The identification of such heterogeneity wouldn't have been possible using the A2a-Cre mouse line. Additionally, the A2a-Cre line could have been useful for the DS but not for the Acb, since the overlap between D2R and A2aR is far from being complete as we have reported in our previous study (Gangarossa et al., 2013 Front Neural Circuit). Finally, the use of A2a-Cre line could also bring major confounds since A2a is highly expressed in endothelial cells.

- for the transcriptional profiling, the authors perform an extensive "functional" analysis of D2R-expressing cells in the D and V striatum. However it seems excessive to claim "that the DS- and Acb-enriched genes have distinct biological functions" based solely on GO-type analyses. Furthermore, there is an even larger leap from this to "these results indicate that D2R neurons have distinct biological functions depending on their anatomical localization within the striatum...."

In agreement with this comment we have tuned down the claims on functional differences on biological functions. Our GO analysis of "Biological Processes" (Figure 6 from the initial version of our manuscript), revealed however that one of the 60 GO terms overlapped between DS and Acb. As a proof of concept, we have illustrated this point by showing the preferential expression of mitochondrial-related genes in the DS, which was confirmed by functional analysis. In the revised version of the manuscript, **Figure 6** showing the three GO analyses has been removed and we only show the GO terms in **Table S5, S6 and S7**.

Minor Points:

- description of D2R LOF might be better understood via ISH as opposed to immuno. As the authors themselves report, the presence of D2R protein in cortical terminals, ChAT interneurons makes the immuno hard to interpret. ISH would show the true percentage of Wfs1+ neurons that have undergone recombination.

We thank the reviewer for his/her remark. We have now performed Basescope assay using a custom-made probe against exon2 of *Drd2* (**Figure 6**) and show clearly the lack of *Drd2* mRNAs in *Wfs1*-positive neurons of D2R-cKO mice. It is important to keep in mind that because of the chromogenic detection accurate quantification cannot be performed.

- what is this non-striatal expression of Wfs1+ cells in cortex? Do they express D2Rs or can they be ignored?

Expression pattern analysis of HA in the cortex of *D2-Ribotag* mice revealed that HA is present in all the cortical layers in both pyramidal and interneurons (unpublished observation). However, no reliable D2R immunostaining could be detected in cortical layer 2/3 pyramidal cells where *Wfs1* is highly expressed.

- "no expression of Wfs1 was detected in CINs...." This is important and should be shown.

We have now included immunostaining and RNAseq data supporting the lack of Wfs1 in CINs as well as in other classes of striatal interneurons.

Reviewer #2:

This manuscript addresses two questions. By using a Wfs1-CreERT in a floxed D2R mouse the authors were able to knockout D2R in a subset of cells expressing this receptor primarily in the Acb region of the ventral striatum. These partial KO mice displayed a phenotype characterized by decrease digging and marble burying as well as increased locomotor activity in response to amphetamine. A separate question concerned the transcript profiles of D2R neurons in the ventral versus dorsal striatum and this was addressed by using a ribosome tagging approach driven by a D2R-Cre crossed with Ribotag.

Critique:

1. These two questions are not really related and I question whether they belong in the same paper. It would have been interesting to determine the transcripts expressed in the subset of D2R neurons that express Wfs1 compared to those that do not but this would require more sophisticated analysis.

We thank the reviewer for his/her comments and suggestions. We have now reorganized the manuscript and performed several additional experiments to better address the issue of unrelated questions. On the other hand, as the reviewer mentioned, comparing the transcripts expressed in D2R+Wfs1+ neurons with D2R+Wfs1- neurons requires state-of-the art intersectional approaches using tools currently not available. Following his/her suggestion, we have performed new experiments to compare the transcripts expressed in Acb D2R+ neurons with Acb Wfs1+ neurons and these data are now included in **Figure 5**.

2. The projections from the Wfs1 neurons were partially traced but the actual circuit downstream of the Wfs1/D2R neurons was not determined. The behavioral changes with this partial KO are interesting but the authors need to follow up with a more detailed mechanistic study to make it meaningful

As suggested by the reviewer, we attempted to identify the circuit downstream of Wfs1/D2R by optogenetic approaches but, unfortunately, mice did not show digging behavior during the entire test –independently of the laser being ON or OFF– (**Figure 1** of this letter). Besides, in the discussion we mention that since D2R-expressing neurons from the Acb only project to the ventral pallidum, the subcircuit Acb-ventral pallidum of Wfs1-positive neurons is likely the one to be involved in dopamine-controlled digging.

On the other hand, with the new reorganization of the manuscript, we show the Drd2-cKO behavioral data as a proof of concept of our deep region-specific molecular analysis serving as a tool to manipulate D2R from specific SPN subpopulations. We believe that assessing a more detailed mechanistic study of the role of D2R from Wfs1+ cells in certain behaviors would make perceive this story even more as two different papers and therefore we think it is beyond the scope of this manuscript.

3. The D2R-Cre/Ribotag study is not properly controlled. It appears that all of the RNA-seq was done on the precipitated polysome fraction without comparison to the inputs for the Acb or dorsal striatum (DS). In the QPCR experiment shown in Fig. 3, the enrichment over total RNA in the D2R neurons appears to be 2-4 fold and the de-enrichment of DIR is about 4 fold. There is significant background expected for these IP experiments and the only way to filter this out is to determine which genes are

significantly enriched in the IP fraction. This would require analyzing the inputs for each experiment by RNA-seq and then filtering out the genes which are not expressed or enriched in D2R neurons

As suggested by the reviewer, we sequenced the input fractions of DS and Acb. To make a proper comparison with the pellet fraction, instead of reusing the RNAseq data from the pellets that we initially sequenced, we generated additional D2R:RiboTag mice and sequenced at the same time the new inputs and pellets. Consequently, in the revised version of the manuscript we have replaced the RNAseq graphs and gene classifications with the new data in Figures 1, 2, 3, S5, S6, S7, S8, and S9. Remarkably, the new RNAseq of the pellet fraction was very similar to the initial RNAseq performed, hence highlighting the reproducibility of our experiments.

We filtered out the genes that are not expressed or enriched in D2R neurons as suggested by the reviewer and plotted this results in Figure 1. Although this analysis allowed us to identify those genes exclusively from D2R neurons that are differentially expressed between DS and Acb, we believe that by filtering the pellet with the input we exclude other relevant genes. Given that the overall gene expression pattern is similar in D1R and D2R cells (results found by Gokce et al, 2016 and us), if a gene has a differential expression between D2R cells from DS and Acb but is equally –or more expressed– in D1R it will be filtered out. Thus, in addition to the filtered data presented in Figure 1, we also broadened our analysis to capture all genes with DS-Acb differential expression regardless of their expression profiles outside D2R neurons. Nevertheless, as suggested by the reviewer, we created Table S1 that includes all the differentially expressed genes after filtering the pellet with the input in our 3 experimental groups: D2R cells from DS, D2R cells from Acb, and Wfs1 cells from Acb.

4. The D2R-Cre was not conditional so any cell that expressed D2R at any time during development would activate the Ribotag in all of the differentiated cells that develop from those progenitors. Is there any literature on the developmental expression of D2R in the embryo?

We thank the referee for his/her comment. Despite the plethora of studies investigating the role of dopamine receptors, little is known regarding the precise developmental expression of D2R within the striatum. The most comprehensive study was recently published by Tinterri et al., (2018) who show the progressive insertion of D2R neurons into the dorsal striatum. No data is available, to our knowledge, concerning the developmental profile of Acb D2R neurons. Future RNAseq analyses at different embryonic stages would certainly constitute a significant step forward this direction.

5. Given the detailed single cell analysis of mRNA expression in the striatum by Gokee et al in 2016 it is unclear how this manuscript adds significantly to that information. The Gokee group did not do an anatomical selection for their study but the Allen Brain ISH allows one to identify where marker genes coexpressed with D2R are localized as the authors show in supplemental Fig3 of this manuscript

Gokce et al (2016) and more recently Saunders et al. (2018) used single-cell approaches to perform unbiased striatal cell type classification. Although informative, these results are however limited to the most expressed genes and will benefit from comparison to the large scale

analysis provided by our study. Importantly, our approach –which allowed the identification of ribosomes-bound mRNAs– revealed that the molecular heterogeneity of striatal D2R neurons cannot be apprehended without taking into account their anatomical localization throughout the dorso-ventral axis of the striatum. Apart from adding new information about the striatal localition of D2R cells genes, we have now included new information regarding the translatomic profile of Wfs1-expressing neurons. Moreover, in contrast to Gokce et al, our study provides functional data, such as the different metabolomic profiles between DS and Acb (preferentially expressed D2R cells genes in DS and increased activity of Complex I) and behavioral data regarding D2R’s role in digging behavior as well as under a psychostimulant effect. Overall, we believe that our study provides a significant contribution to the basal ganglia field by deciphering novel molecular markers that define discrete striatal subpopulations and uncovering the role of D2R from one of these subpopulations in specific motor traits.

Reviewer #3 :

In their manuscript, Puighermanal et al. describe a collection of experiments related to D2R-expressing neurons in the dorsal striatum/accumbens. There are a number of results, which I will try to summarize here:

- 1. When D2R expression was reduced in wfs1 neurons (found mostly in the accumbens), mice exhibit specific behavioral deficits in digging and marble burying, but not in any other of a number of behavioral tasks. These same mice also exhibited exaggerated amphetamine-induced locomotor responses.*
- 2. The authors used D2R-Cre:RiboTag mice to isolate ribosome-bound RNAs from D2R-expressing neurons in either the dorsal striatum or the accumbens. A number of genes were found that were preferentially expressed in one compartment or the other which were then validated with qRT-PCR and in some cases, with immunostaining.*
- 3. Using available data bases, the authors then classified genes identified as exhibiting a compartment-specific gradient.*
- 4. Finally, the authors performed gene ontology analysis to further analyze their data, but I found this section hard to follow.*

Overall, this manuscript seemed like a series of observations, not necessarily related, combined into a single manuscript. The behavioral results in particular seemed quite disconnected from the rest of the manuscript, and I would suggest that the behavior and gene expression work be separated in any subsequent manuscripts. The behavioral results are interesting, but preliminary.

We thank the reviewer for his/her remarks. In the revised version of the manuscript we have now changed the organization of the paper in a way that we first show the novel region-specific markers we identified by RNAseq, immunofluorescence, and ISH that may serve as a tool to target and manipulate genetically-defined striatal subpopulations. As a proof of concept, we then chose Wfs1-expressing cells, we characterized this novel subpopulation by revealing its transcriptome profile and connectivity, and finally we selectively deleted D2R from Wfs1+neurons in adult mouse to uncover the specific role of D2R in behavior. We hope that with this new organization all the data can be perceived as one story.

Reviewers' Comments:

Reviewer #1:

Remarks to the Author:

In my previous review, I thought by "not focusing on one specific issue, this manuscript does not provide sufficient novel insight to warrant publication in Nature Communications". This was a conclusion seemingly reached by all 3 reviewers. I acknowledge that the authors have performed a significant amount of work in trying to address several of my major comments. Nevertheless, I still find this work very borderline for publication as is. There remain major issues:

1. the title does not accurately describe the main take-home: this likely reflects a broader issue with putting these two disparate stories together. Currently, the balance throughout the manuscript feels skewed in the opposite direction of the title with the stronger content/more thorough analysis coming from the transcriptional analysis side. At the very least, the title should reflect the diversity of D2R+ cell types in the D-V axis, as this is the most novel and most solid outcome.
2. the Wfs1+ sub-circuit remains unclear on several levels – a. why pick this gene as a follow-up to all of the transcriptional analysis? it clearly has a D-V bias but its expression is more biased towards D1R+ SPNs while this study has been framed in terms of D2R+ neuronal populations; b. the contributions of this circuit to behavior remain relatively unclear – is this really a digging specific circuit? if the optogenetic/chemogenetic manipulations have no impacts on behavior, doesn't this suggest that creating circuits based on single gene profiles (that cut across other known subtype indicators) is perhaps arbitrary and not revealing any meaningful underlying circuit architecture?
3. the section about ChINs is yet another angle that is distracting and unclear. I remain uncertain how the authors distinguish ChIN vs iSPN transcripts given that they are mixed together because of the D2-Cre allele crossed to the RiboTAG. it seems like they are using pre-existing databases - if so, doesn't it seem weird (and circular) to use other genetically-defined ChIN transcriptional datasets to uncover the ChIN-specific genes in your data? if the authors wanted to make claims about ChIN, the appropriate genetic lines should have been used.
4. regarding the idea of a digging circuit – it seems insufficient to say that the phenotype doesn't represent an inability to continue started behaviors based on no changes in repetitive grooming or rotarod learning. would less grooming (given how rare these events are) have been convincing evidence for an inability to continue initiated behaviors? the functional relevance of this wfs1+ still seems underdeveloped.

as in my first review, i still view the above as major issues. beyond this, the data is of high quality both in terms of the experimental paradigms, statistical analysis and representation. Wherever published, the profiling will make an excellent resource.

Reviewer #2:

Remarks to the Author:

The authors have added a considerable amount of new data on their RiboTag analysis of D2R neurons in the Acb and DS and have also added experiments using the Wfs1 Cre to identify transcripts specifically transcribed in those cells. In all experiments the authors have now included an analysis of the input so that one can determine the relative enrichment in each Cre expressing cell type. This is valuable information and allows them to filter for genes which have significant expression in the neuron of interest.

The reason for doing this is that the immunoprecipitation of polysomes like all IP experiments will

bring down some background polysomes that are not HA-labeled. This is quite evident in the author's data when looking at IP/input values for glial cell markers for example in Fig. 1b or Fig. 4d. However, this de-enrichment can be quite variable as the authors and others have shown so it is difficult to filter out the background precisely. As an example, the authors go on to analysis their data without filtering out the background and show that Ttr is the most highly "enriched" transcript when comparing DS to Acb D2R transcripts (Fig. 1J). However, based on Table 1 Ttr which is almost exclusively expressed in the choroid plexus has an IP/input value of 0.018 in Acb and 0.21 in DS. The DS is in proximity to the lateral ventricle where there is extremely high expression of Ttr which suggests that the higher value in the DS is just background.

Transferrin (Trf) which is a marker for oligodendrocytes has IP/input values of 0.12 (Acb) and 0.13 (DS) and the microglial marker Aif1 has IP/input values of .34 (Acb) and .41 (DS). Therefore, the author's analysis when just comparing the IP pellets with each other is problematic because it is potentially just comparing background levels in the individual IPs. It is not necessary to filter for an IP/input of 1.0 or greater but values should at least be greater than that determined from the control genes that are known to be glial or non-D2R expressing neurons.

The author's should also be more precise in describing what they mean when they use the term enrichment. In some results enrichment is describing the IP/input for one anatomical structure which seems appropriate but in other cases enrichment is used to describe differential expression between two anatomical structures (IP in Acb vs IP in DS). This is quite different and should be described more clearly.

The authors attempted some very nice experiments to further explore the role of Wfs1/D2R neurons in the behavioral digging and locomotor changes observed and it is unfortunate that both the optogenetic and chemogenetic attempts did not yield more interpretable data. Nevertheless, the behavior remains an interesting observation.

Responses to Referees

We are thankful for the referees' evaluation of our manuscript and for their positive comments and constructive feedback. We are delighted that our work was described as "*of high quality both in terms of the experimental paradigms, statistical analysis and representation*" and that "*the profiling will make an excellent resource.*"

We provide a point-by-point discussion of the referee's comments below, and hope they find our revised study suitable for publication in *Nature Communications*.

Reviewer #1:

In my previous review, I thought by "not focusing on one specific issue, this manuscript does not provide sufficient novel insight to warrant publication in Nature Communications". This was a conclusion seemingly reached by all 3 reviewers. I acknowledge that the authors have performed a significant amount of work in trying to address several of my major comments. Nevertheless, I still find this work very borderline for publication as is. There remain major issues:

We appreciate that the reviewer acknowledges we performed "*a significant amount of work in trying to address several of my major comments*".

1. the title does not accurately describe the main take-home: this likely reflects a broader issue with putting these two disparate stories together. Currently, the balance throughout the manuscript feels skewed in the opposite direction of the title with the stronger content/more thorough analysis coming from the transcriptional analysis side. At the very least, the title should reflect the diversity of D2R+ cell types in the D-V axis, as this is the most novel and most solid outcome

We have now re-structured the manuscript to better emphasize the transcriptional analysis. We have now clearly stated that the identification of hundreds of novel region-specific molecular markers may be used as tools to target selective striatal subpopulations, and specified that the part of the study related to *Wfs1* subcircuit serves as a proof-of-concept illustrating this point. Title and abstract have been therefore modified accordingly to reflect that our work would provide valuable resource to the field.

2. the Wfs1+ sub-circuit remains unclear on several levels – a. why pick this gene as a follow-up to all of the transcriptional analysis? it clearly has a D-V bias but its expression is more biased towards D1R+ SPNs while this study has been framed in terms of D2R+ neuronal populations;

As stated in the revised version of the manuscript we picked this marker because it displayed one of the most segregated expression patterns along the dorsal and ventral axis. Therefore, the differential expression of *Wfs1* between the DS and Acb (top gene with bias expression) was used to validate our transcriptomic data. In addition, genetic tools were also available to study

this cell population, which allowed us to generate temporally-controlled conditional D2R knock-out mice. The molecular characterization of Acb Wfs1 neurons using *Wfs1-Ribotag* mice provided in the revised version of the study revealed that transcripts expressed in both dSPNs (*Drd1, Pdyn, Tac1, Chrm4*) and iSPNs (*Drd2, Adora2a, Penk, Gpr6*) were enriched in Wfs1 neurons providing a rationale to study the functional role of D2R within this neuronal population.

b. the contributions of this circuit to behavior remain relatively unclear – is this really a digging specific circuit? if the optogenetic/chemogenetic manipulations have no impacts on behavior, doesn't this suggest that creating circuits based on single gene profiles (that cut across other known subtype indicators) is perhaps arbitrary and not revealing any meaningful underlying circuit architecture?

Our chemogenetic experiments (now included in the manuscript as **Supplementary Figure 11**) suggest that activation of whole Wfs1-positive subcircuit (including both dSPN and iSPN subpopulations) had no impact on digging behavior. These results clearly point out the limitation of studying circuits based on single gene profiles. In the present case, the overcoming of this limitation requires the development of new genetic tools (*Drd2-flp* or *Drd1a-flp* mice). Such intersectional approaches will be crucial to determine the functional role of Wfs1/D2R- vs Wfs1/D1R-populations. We have included this into our discussion.

3. the section about ChINs is yet another angle that is distracting and unclear. I remain uncertain how the authors distinguish ChIN vs iSPN transcripts given that they are mixed together because of the D2-Cre allele crossed to the RiboTAG. it seems like they are using pre-existing databases - if so, doesn't it seem weird (and circular) to use other genetically-defined ChIN transcriptional datasets to uncover the ChIN-specific genes in your data? if the authors wanted to make claims about ChIN, the appropriate genetic lines should have been used

We would like to thank reviewer for the suggestion. We have now shortened the section related to the CINs. These results are however important to include because they highlight the sensitivity of the Ribotag approach compared to the D2-bacTRAP, in which CINs markers were not detected. Our cross-analysis with recent single-cell RNAseq studies provided important new insights regarding the molecular heterogeneity among CINs and also illustrated the complementarity for the two approaches. Finally, we believe this section is important because we provided here a molecular explanation for the differential effect of D2R activation of CINs of the DS vs Acb (see paper of Chumma et al., 2014).

4. regarding the idea of a digging circuit – it seems insufficient to say that the phenotype doesn't represent an inability to continue started behaviors based on no changes in repetitive grooming or rotarod learning. would less grooming (given how rare these events are) have been convincing evidence for an inability to continue initiated behaviors? the functional relevance of this wfs1+ still seems underdeveloped

As suggested by the reviewer and editor, we have now re-structured the manuscript to better emphasize the resource nature of current work. Title and abstract have been therefore modified accordingly to reflect this change. In addition, we tuned down the detailed behavioral analysis

of Wfs1+ neuron in digging behavior, but rather emphasize that the study related to Wfs1 subcircuit serves as a proof-of-concept validating our findings.

Reviewer #2 (Remarks to the Author):

The authors have added a considerable amount of new data on their RiboTag analysis of D2R neurons in the Acb and DS and have also added experiments using the Wfs1 Cre to identify transcripts specifically transcribed in those cells. In all experiments the authors have now included an analysis of the input so that one can determine the relative enrichment in each Cre expressing cell type. This is valuable information and allows them to filter for genes which have significant expression in the neuron of interest.

We appreciate that the reviewer acknowledges we added “*a considerable amount of new data on their RiboTag analysis of D2R neurons in the Acb and DS and have also added experiments using the Wfs1 Cre to identify transcripts specifically transcribed in those cells*”

The reason for doing this is that the immunoprecipitation of polysomes like all IP experiments will bring down some background polysomes that are not HA-labeled. This is quite evident in the author's data when looking at IP/input values for glial cell markers for example in Fig. 1b or Fig. 4d. However, this de-enrichment can be quite variable as the authors and others have shown so it is difficult to filter out the background precisely. As an example, the authors go on to analysis their data without filtering out the background and show that Ttr is the most highly "enriched" transcript when comparing DS to Acb D2R transcripts (Fig. 1J). However, based on Table 1 Ttr which is almost exclusively expressed in the choroid plexus has an IP/input value of 0.018 in Acb and 0.21 in DS. The DS is in proximity to the lateral ventricle where there is extremely high expression of Ttr which suggests that the higher value in the DS is just background.

We agree with the reviewer that it is difficult to filter out the background precisely since there is no general rule of where to set a threshold. As he/she initially suggested, we performed additional RNAseq analyses comparing the pellet fractions (after the HA immunoprecipitation) with the input fractions (containing mRNAs from all cell types) and filtered out the genes that are not expressed or enriched in D2R neurons (**Figure 1 and Supplementary Table 1**). Although this analysis allowed us to identify those genes exclusively from D2R neurons that are differentially expressed between DS and Acb, we believe that by filtering the pellet with the input we exclude other relevant genes. Given that the overall gene expression pattern is similar in D1R and D2R cells (results found by Gokce et al, 2016 and us), if a gene has a differential expression between D2R cells from DS and Acb but is equally –or more expressed– in D1R it will be filtered out. Thus, in addition to the filtered data, we also broadened our analysis to capture all genes with DS-Acb differential expression regardless of their expression profiles outside D2R neurons.

As pointed out by the reviewer, in Fig. 1j we did not filter the data and we found a more abundant presence of *Ttr* in D2 cells from the DS compared to Acb. Although we cannot certainly exclude that this value is background, we believe that it reflects the *Ttr* expression in D2R cells. First, although early studies –mainly using Northern blotting– suggested that the choroid plexus was the only site in the brain where *Ttr* expression took place, later studies using microarrays of RNA from different brain regions showed mouse strain and regional differences of *Ttr* transcript expression, and more recent studies suggested that there is a very low level of neuronal TTR synthesis (for review, see Li and Buxbaum, 2011). In agreement, the number of *Ttr* reads in our RNAseq is only 24, confirming its low expression levels in neurons (Supplementary Table 1). Second, the log2foldchange between D2R DS and Acb pellets is 5.9 ($p < 0.001$), whereas it is only 2.8 between D2R DS and Acb inputs ($p = 0.013$)

(Supplementary Table 1). We believe that if the log2foldchange 5.9 was a mere product of the background, the log2foldchange of the input would also be higher. Besides, in addition to the non-filtered data we also provide all our data filtering the pellets with the inputs to totally exclude a background effect.

Transferrin (Trf) which is a marker for oligodendrocytes has IP/input values of 0.12 (Acb) and 0.13 (DS) and the microglial marker Aif1 has IP/input values of .34 (Acb) and .41 (DS). Therefore, the author's analysis when just comparing the IP pellets with each other is problematic because it is potentially just comparing background levels in the individual IPs. It is not necessary to filter for an IP/input of 1.0 or greater but values should at least be greater than that determined from the control genes that are known to be glial or non-D2R expressing neurons.

We agree with the reviewer's comment and this is actually why we performed 2 analyses in parallel in our additional RNAseq: filtering and not filtering. However, we do not understand where those values come from. In Supplementary Table 1, as expected, both *Trf* and *Aif1* are de-enriched in all pellet fractions compared to the input fractions in both DS and Acb:

	D2-RiboTag_Acb_PELLETT vs D2-RiboTag_Acb_INPUT		D2-RiboTag_DS_PELLETT vs D2-RiboTag_DS_INPUT		Wfs1-RiboTag_Acb_PELLETT vs Wfs1-RiboTag_Acb_INPUT	
gene	log2foldchange	padj	log2foldchange	padj	log2foldchange	padj
Trf	-3.043491594	0	-2.889831652	0	-2.777933885	0
Aif1	-1.539654543	9.05E-14	-1.288281893	8.80E-10	-1.018869232	9.91E-07

As stated in the text, we think that by using only the filtered data, we might exclude many differentially expressed genes between D2R from the DS and D2R from the Acb –regardless of their expression profiles outside of D2R neurons– and hence we also analyzed the data without prefiltering.

We thank the reviewer for his suggestion of sequencing the inputs and filtering the pellets. Now, all the analyses (filtering and not filtering) are provided in this study as well as the raw data so that any user can filter for a given IP/input ratio as suggested by the reviewer.

The author's should also be more precise in describing what they mean when they use the term enrichment. In some results enrichment is describing the IP/input for one anatomical structure which seems appropriate but in other cases enrichment is used to describe differential expression between two anatomical structures (IP in Acb vs IP in DS). This is quite different and should be described more clearly.

We have now clarified this issue in the results section.

The authors attempted some very nice experiments to further explore the role of Wfs1/D2R neurons in the behavioral digging and locomotor changes observed and it is unfortunate that both the optogenetic and chemogenetic attempts did not yield more interpretable data. Nevertheless, the behavior remains an interesting observation

Thank you! We appreciate his/her positive comments.